# CUL4A-DDB1-DCAF10 is an N-recognin for N-terminally acetylated Src kinases

Nora Kremer [1,2,5], Franziska Mueller [3,5], Hang Nguyen [1,2], Louisa Schulz [1,2], Tanja Popp[1], Elena Artes[1,2], Julian Wolters [1], Michael Renner [1,2], Ingrid Vetter[3], Stefano Maffini[3], Maria S. Robles [2], Andrea Musacchio [3,4] & Tanja Bange [1,2] ✉

Co-translational N-terminal modifications such as methionine excision, acetylation, and myristoylation govern protein stability, localization, and folding. Disruption can expose N-terminal degrons that trigger ubiquitin-mediated degradation, safeguarding the proteome. N-terminal acetylation usually protects proteins from degradation, but can also promote it through the Ac/N-degron pathway. Src-family kinases (SFKs), signaling enzymes implicated in tumorigenesis, require N-terminal myristoylation for function. Using peptide pull-downs, mass spectrometry, and AlphaFold 3 predictions, we identify DCAF10 as the E3 ligase substrate receptor for alternatively N-terminally acetylated SFKs. Combining siRNA-mediated knockdown and CRISPR/Cas9-mediated knockout of endogenous Lyn with inducible Lyn-GFP variants confirms that DCAF10 regulates SFK levels by recognizing an N-terminal acetylated glycine residue. In vitro, a CUL4A-DDB1-DCAF10 complex ubiquitinates N-terminally acetylated SFKs. Thus, we define a novel N-degron pathway that monitors replacement of myristoylation by acetylation and activates degradation of SFKs upon acetylation. This mechanism may extend to other N-terminally myristoylated proteins beyond SFKs.

As nascent proteins emerge from the ribosome, their N-termini often undergo co-translational modifications, including initiator methionine (iMet) excision, N-terminal (Nt) acetylation, or N-myristoylation[1–4]. These modifications influence protein function, localization, and stability. Failure to properly modify proteins can lead to misfolding and degradation via the ubiquitin-proteasome system, which targets approximately 12–15% of nascent polypeptides[5].

Protein degradation is largely mediated by E3 ligases, which recognize degrons, specific sequence motifs that mark proteins for ubiquitination. Degrons at the N-terminus, known as N-degrons, are recognized by specialized E3 ligases called N-recognins[6]. N-degron pathways can function constitutively or be conditionally triggered.

N-degrons are typically short (1–4 amino acids) with the Nt-amino acid or its modification serving as the primary recognition site[6–8]. Nt-acetylation, catalyzed by the N-terminal acetyltransferase A (NatA), generally stabilizes proteins by masking potential N-degron signals[9,10]. NatA acetylates small Nt-residues (Ala, Ser, Thr, Val, Gly, and Cys) after iMet removal. The NatA substrate context is well defined, and proteome mapping plus mechanistic studies show that loss of NatA unmasks conserved non-Ac/N-degrons, accelerating substrate turnover[9–12]. When NatA acetylation is absent, free Ala and Ser N-termini can be recognized by IAP-family E3s, and free Gly, Ala, Ser, Thr, and Cys by the CUL2 adapters ZER1/ZYG11B[11,13]. In parallel, failure of N-myristoylation leaves Nt-Gly unmodified and targets substrates to

[1]Department of Medicine II, LMU University Hospital, Munich, Germany. [2]Faculty of Medicine, Institute of Medical Psychology and Biomedical Center (BMC), LMU Munich, Munich, Germany. [3]Department of Mechanistic Cell Biology, Max Planck Institute of Molecular Physiology, Dortmund, Germany. [4]Faculty of Biology, Centre for Medical Biotechnology, University of Duisburg-Essen, Essen, Germany. [5]These authors contributed equally: Nora Kremer, Franziska Mueller. ✉e-mail: tanja.bange@med.uni-muenchen.de

the Gly/N-degron pathway via the same CUL2 axis[14]. NatC similarly buffers iMet-retained Met-Φ (hydrophobic) N-termini, and its loss exposes these starts to UBR4–KCMF1/UBR1/UBR2, promoting degradation[15]. Together, these small-residue branches are termed the GASTC/N-degron pathway[6]. Cys can also engage the classical Arg/N-degron branch when oxidized by ADO (2-aminoethanethiol dioxygenase), forming an entry point for arginylation and degradation[16]. Finally, Nt-acetylation can also enable Ac/N-degron-mediated turnover —recognized by Doa10 (yeast) and MARCH6/TEB4 (mammals), linked to NOT4 during translation—highlighting that acetylation masks destabilizing N-termini when present and can create degrons in a sequence- and context-dependent manner[17–20].

Proteins with Nt Met-Gly sequences, such as the SFKs (including Src, Fyn, Yes, Lyn, Hck, Lck, Fgr, and Blk in humans), present a unique case. Following iMet excision, their N-termini may remain free, undergo Nt-acetylation by NatA, or N-myristoylation by the N-myristoyltransferases 1 or 2 (NMT1/2). These modifications are thought to act on overlapping substrates and may influence localization, signaling, and protein stability[1,21]. N-myristoylation is essential for membrane localization, activity and oncogenic signaling of SFKs[22,23]. Without this modification, the exposed free Gly residue becomes a target for the Gly/N-degron pathway[14]. However, an Nt-acetylated variant of Src has also been observed, although its biological significance remains unclear[21].

The expanding network of N-degron pathways highlights the intricate interplay between co-translational modifications, protein quality control, and targeted protein degradation. Here, we show that when N-myristoylation of SFKs is replaced by NatA-mediated Nt-acetylation, the acetylated state functions as a bona fide N-degron. We identify CUL4A-DDB1-DCAF10 as the cognate N-recognin for this pathway. Thus, N-degron systems do not merely remove proteins lacking required modifications; they can also read out alternative, physiologically accessible Nt-states to fine-tune kinase homeostasis.

## Results

### DCAF10 binds acetylated N-terminal glycine residues

We previously employed peptide pull-downs combined with quantitative mass spectrometry (MS) to identify N-recognins of proteins lacking physiological Nt-acetylation[11,24]. Beads presenting 10-residue synthetic Nt-peptides of proteins of interest in their free (Nt-free) or modified (e.g., Nt-acetylated) versions were incubated with lysates from HeLa cells, followed by MS-based quantification of retained binders (Fig. 1a). This identified the inhibitor of apoptosis proteins (IAPs) BIRC2, BIRC3, BIRC6 and XIAP as N-recognins for N-termini lacking physiological Nt-acetylation[11].

To test the general applicability of this pull-down approach on a well-defined N-degron system, we examined whether it could recover known components of the CUL2 Gly/N-degron pathway, in which the substrate adapters ZYG11B and ZER1 recognize proteins exposing a free Nt Gly upon loss of N-myristoylation[14]. We therefore selected five proteins whose initial Gly is normally myristoylated, including Lyn, Src, Fyn, Yes, and NDUFAF4. Additionally, we included five proteins whose Nt Gly is subject to a diverse set of Nt-processing events, including GNAI3 (N-myristoylated), ARF1 (N-myristoylated and Nt-acetylated), EF1B and TMEM97 (uncharacterized N-termini), and THOC7 (Nt-acetylated by NatA)[25–27].

Peptide pull-downs using the Nt-free and Nt-acetylated versions of the aforementioned Nt peptides revealed ZYG11B binding only to the Nt-free peptide of TMEM97, likely reflecting the transient nature of ZYG11B/ZER1 interactions with their target proteins (Supplementary Fig. 1a). However, rather than broadly recovering ZYG11B/ZER1, we observed robust binding of the DDB1- and CUL4A-associated factor 10 (DCAF10/WDR32) specifically to the Nt-acetylated versions of nine out of ten Gly-starting peptides (Fig. 1b–d; Supplementary Fig. 1a–f; Supplementary Data 1, 2 and Supplementary Table 1). The only exception

was Nt-acetylated THOC7, which did not bind DCAF10 significantly (Fig. 1e). Repeating pull-down experiments with mouse liver lysates confirmed DCAF10 binding to the Nt-acetylated Lyn peptide, whereas the Nt THOC7 peptides did not show any binding (Supplementary Fig. 1g, h and Supplementary Data 2).

The DCAF family includes approximately 60 members in humans, primarily functioning as substrate receptors in cullin-RING E3 ubiquitin ligase complexes and thus mediating targeted protein degradation[28,29]. DCAF10 may be part of the CUL4A-DDB1-DCAF10 E3 ligase complex (Fig. 1f)[30]. Among the nine DCAF10-binding peptides, seven correspond to proteins known to be N-myristoylated, while two (TMEM97 and EF1B) have not been characterized as such. Further pull-downs using Nt-acetylated and N-myristoylated versions of Lyn, Fyn, and Src peptides confirmed that DCAF10 selectively recognizes the acetylated form of their N-termini (Fig. 1g; Supplementary Fig. 1i–k; Supplementary Data 1, 2 and Supplementary Table 1).

In summary, these pull-down experiments reveal that the E3 ligase substrate receptor DCAF10 preferentially binds Nt-acetylated glycine (Ac-Gly) of Gly-starting proteins, particularly those that are typically N-myristoylated or have uncharacterized N-termini. On the other hand, THOC7, whose N-terminus is robustly Nt-acetylated in cells, does not bind DCAF10. Notably, previous peptide pull-downs using Ala- or Ser-starting peptides never identified DCAF10 as a binding partner for acetylated peptides, suggesting a specific preference for Gly-starting N-termini[11]. Furthermore, additional features may increase the specificity of DCAF10 for certain Gly-starting proteins. A sequence logo of the DCAF10-binding peptides reveals, in addition to an invariant Ac-Gly at position 1, a preference for serine and lysine at positions 6 and 7, respectively (Fig. 1h).

### Acetylated N-termini of SFKs interact directly with DCAF10

All four tested acetylated Nt peptides from SFKs (Lyn, Fyn, Src, and Yes) bound to DCAF10 in the initial pull-downs, suggesting a mechanism relevant to this protein family. We therefore set out to investigate whether DCAF10 directly recognizes Ac-Gly-starting peptides. While SFK N-termini are typically N-myristoylated, there is evidence that Src is strongly (~82%) but not fully myristoylated, and Nt-acetylated Src has been identified in human tissue[21,31].

We used AlphaFold 3 (AF3[32]), which can model interactions with small molecules, to predict interactions between Nt-peptides (aa 2–10) and DCAF10 (aa 120–559, UniprotID: Q5QP82). Nt-acetylated peptides from Lyn, Fyn, and THOC7 were predicted to bind deeply into the β-propeller pocket of DCAF10, with strong pLDDT scores (78–80, pLDDT > 70 are considered reliable) (Fig. 2a–d; Supplementary Figs. 2a and 3a–c left panels). Predictions using the first 39 residues (aa 2–40) showed highly similar binding modes inside the DCAF10 pocket, whereas the more C-terminal residues varied considerably between predictions (Supplementary Figs. 2b–e and 3a–c middle and right panels). In contrast, unmodified peptides docked on DCAF10 without occupying the acetyl-binding pocket, while the myristoylated versions were not predicted to bind (Fig. 2e, f and Supplementary Fig. 2f–i). Thus, AF3 appears to recognize the acetyl group's unique properties as critical for binding. The DCAF10 pocket and Ac-Gly interacting residues of DCAF10 (Phe172, Asn215, Lys257, Ile475, and Glu477) are highly conserved among species (Supplementary Fig. 2j–o). Additionally, Lyn and Fyn residues downstream of Gly (Cys3-Ile4-Lys5, Lyn; Cys3-Val4-Gln5, and Fyn) also interact with highly conserved DCAF10 residues (Met302, Phe273, Trp255, and Tyr474) (Supplementary Fig. 2l). Notably, only the first 3–4 amino acids show significant pLDDT and PAE scores (indicators of AF prediction quality), suggesting that sequence specificity is primarily limited to this region (Supplementary Fig. 3a–c). AF3 unexpectedly predicted pocket engagement for acetylated THOC7 despite its lack of binding in pull-downs. We note that the negatively charged pocket environment may repel acidic residues (Asp6-Asp7-Glu8) near the THOC7 N-terminus, potentially explaining

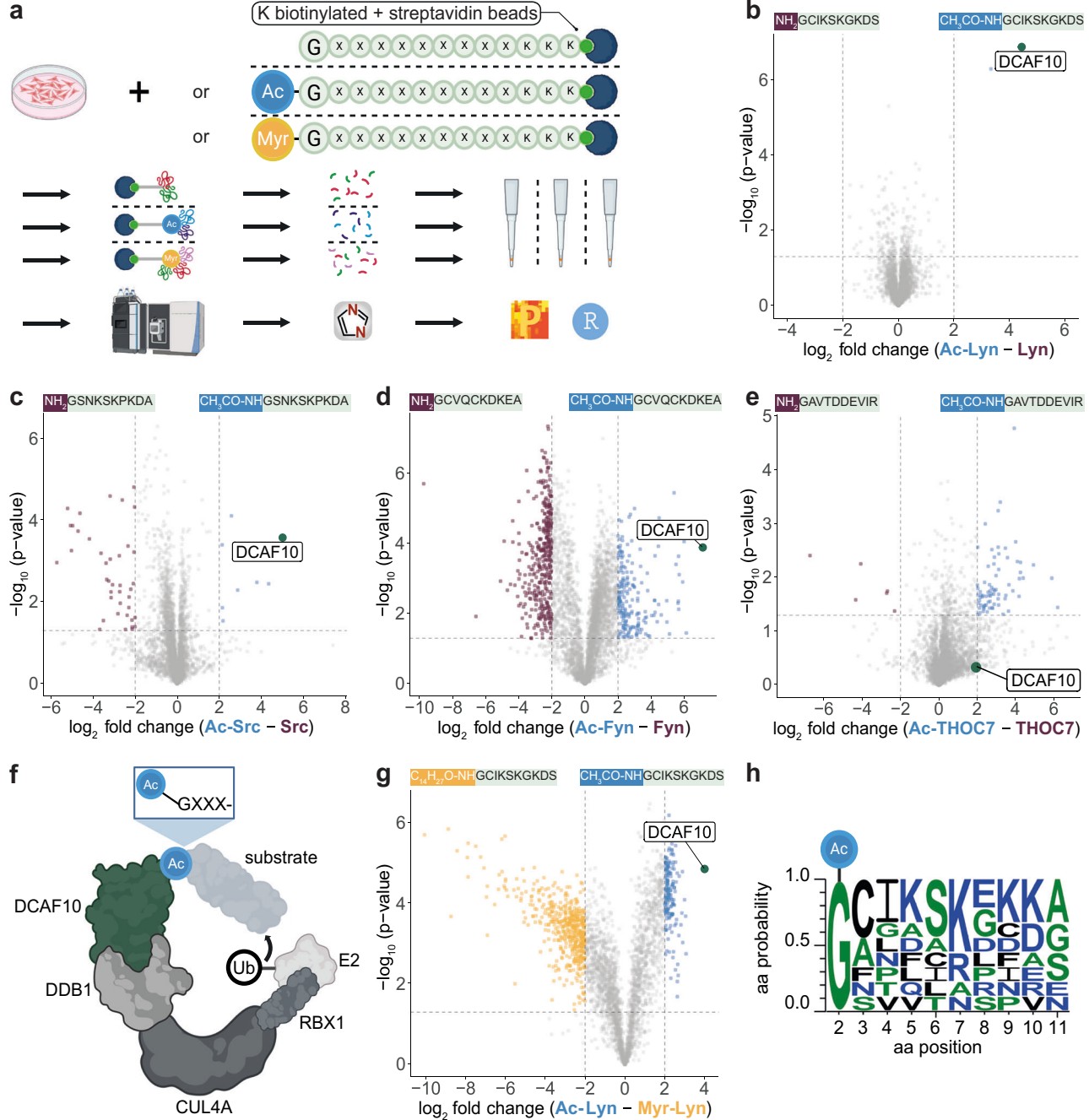

**Fig. 1 | DCAF10 binds acetylated N-terminal glycine residues. a** Schematic of peptide pull-downs combined with quantitative mass spectrometry (MS). Ac acetylation; Myr myristoylation; G glycine; x any amino acid; K lysine. Created in BioRender. Kremer, N. (2025) https://BioRender.com/fxzaxz0. **b–e** Volcano plots comparing binding partners of free (Nt-free) and acetylated (Nt-Ac) N-termini (aa 1–10 + KKK-biotin; Supplementary Table 1) (*n* = 3 independent biological replicates). **b** Lyn; **c** Src; **d** Fyn; **e** THOC7. The −log10 adjusted *p*-value (two-sided Student's *t* test with permutation-based multiple-testing correction; y-axis) is plotted against the log2 fold change (x-axis). Threshold for significance: −log10 *p*-value ≥ 1.3 (*p* ≤ 0.05), log2 fold change ≤ −2 or ≥ 2. DCAF10 is marked in green, significant Nt-

free binders in purple, and significant Nt-Ac binders in blue. **f** Schematic of the CUL4A-DDB1-DCAF10 E3 ligase complex including an Ac-Gly-starting substrate, the catalytic subunit RBX1, E2, and ubiquitin. Created in BioRender. Kremer, N. (2025) https://BioRender.com/0whobwz. **g** Volcano plot comparing Nt-Ac and myristoylated (Nt-Myr) N-termini of Lyn. The −log10 adjusted *p*-value (two-sided Student's *t* test with permutation-based multiple-testing correction) is plotted against the log2 fold change, as in (**b–e**). Threshold for significance: −log10 *P* value ≥ 1.3 (*P* ≤ 0.05), log2 fold change ≤ −2 or ≥2 (*n* = 3 independent biological replicates). **h** Sequence logo of nine N-terminal sequences significantly binding to DCAF10 (including six represented in Supplementary Fig. 1).

the lack of binding in our experiments (Figs. 1e; 2d and Supplementary Figs. 1h; 2e). Charge complementarity (or lack thereof) is not explicitly modeled by AF3 in these peptide–protein predictions.

To confirm this hypothesis, we exchanged the positively charged segment Lys5-Ser6-Lys7 of Lyn with the negatively charged segment Thr5-Asp6-Asp7 of the THOC7 sequence (Lyn^swap) and vice versa

(THOC7^swap) and performed peptide pull-downs as before (Fig. 2g). Although the THOC7 acidic patch lies at residues 6–8, we engineered swaps at 5–7 to modulate local charge within the motif's specificity window (Fig. 1h), where Lyn carries Lys at positions 5 and 7. In the WT comparison, Ac-Lyn bound DCAF10 ~57-fold more strongly than Ac-THOC7, whereas after swapping positions 5–7, the preference reversed

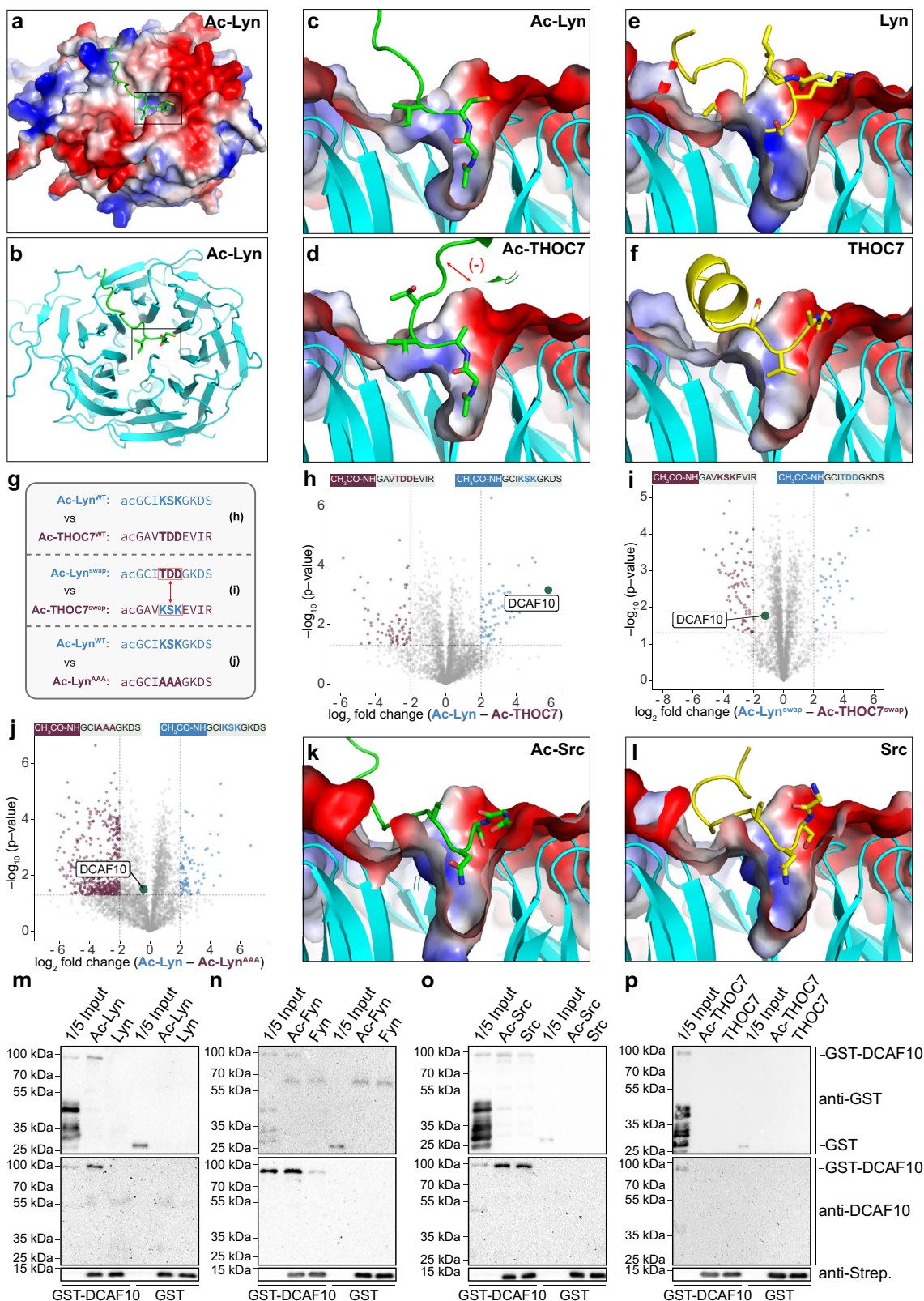

by ~133-fold (log₂ difference Ac-Lyn vs Ac-THOC7: 5.83; log₂ difference Ac-Lyn$^{swap}$ vs Ac-THOC7$^{swap}$: −1.23). An Ala5-Ala6-Ala7 mutant of Lyn (Lyn$^{AAA}$) bound similarly to Lyn$^{WT}$ (log₂ difference: −0.41) (Fig. 2h–j and Supplementary Data 3). Direct comparisons of Ac-Lyn versus Ac-Lyn$^{swap}$ and Ac-THOC7 versus Ac-THOC7$^{swap}$ confirmed, as expected, significantly reduced and increased binding, respectively, for the mutant peptides (Supplementary Fig. 3e, f and Supplementary Data 3).

These data reconcile AF3 and peptide pull-down results, showing that Ac-Gly is a binding anchor, whereas acidic residues disfavor binding.

Returning to the AF3 predictions, the Src peptide was placed largely outside the pocket irrespective of acetylation, with lower, yet still significant, pLDDT and PAE scores (pLDDT 74.9) (Fig. 2k, l; Supplementary Figs. 2p, q and 3d).

**Fig. 2 | Acetylated N-termini of Src-family kinases interact with DCAF10.**
**a**–**f** AlphaFold 3 predictions of DCAF10 (aa 120–559; UniprotID: Q5QP82) bound to Nt-peptides. **a** DCAF10 with Nt-Ac Lyn; **b** cartoon of the DCAF10 β-propeller with Nt-Ac Lyn (black boxes indicate inlets shown in (**c**–**f**, **k**, **l**); **c**–**f** cross-sections of the DCAF10 β-propeller, highlighting the deep tunnel. **c** Nt-Ac Lyn; **d**, Nt-Ac THOC7 (red arrow indicates potential repulsion between acidic side chains (Asp6-Asp7-Glu8) and the acidic DCAF10 patch; Supplementary Fig. 2e); **e** Nt-free Lyn; **f** Nt-free THOC7. DCAF10 surface is colored by electrostatic potential (red = negative, blue = positive, white = neutral or hydrophobic). Nt-Ac peptides: green, Nt-free peptides: yellow. **g** Schematic Lyn and THOC7 swapped and mutant peptides. Created in BioRender. Müller, F. (2025) https://BioRender.com/p80a894. Volcano plots comparing binding partners for **h** Ac-Lyn^WT vs. Ac-THOC7^WT; **i** Ac-Lyn^swap vs Ac-

THOC7^swap; **j** Ac-Lyn^WT vs. Ac-Lyn^AAA. The −log10 adjusted p-value (two-sided Student's t test with permutation-based multiple-testing correction; y-axis) is plotted against the log2 fold change (x-axis). Threshold for significance: −log10 p-value ≥ 1.3 ($p \leq 0.05$), log2 fold change ≤ −2 or ≥ 2. DCAF10 is marked in green, significant binding partners in purple and in blue ($n = 3$ independent biological replicates). **k**, **l** Cross-sections of the DCAF10 β-propeller, highlighting the deep tunnel. **k** Nt-Ac Src; **l** Nt-free Src. **m**–**p** GST-DCAF10 or GST alone incubated with Nt-peptides bound to streptavidin beads. Nt-Ac and Nt-free peptides of **m** Lyn, **n** Fyn, **o** Src, and **p** THOC7. Western blots (WBs) against GST, DCAF10, and streptavidin. ($n = 3$ independent binding assays, quantification in Supplementary Fig. 4b). Source data are provided as a Source data file.

To confirm direct interactions between peptides and DCAF10, we performed pull-down assays using bead-bound Nt-peptide variants incubated with purified full-length GST-DCAF10 or GST as a negative control (Fig. 2m–p and Supplementary Fig. 4a). Nt-acetylated Lyn and Fyn peptides bound robustly to GST-DCAF10, whereas Nt-free variants bound only weakly (Lyn Nt-free reduction compared to Nt-Ac: 88.5 ± 14.7%; Fyn Nt-free reduction compared to Nt-Ac: 82.5 ± 10.8%) (Fig. 2m, n and Supplementary Fig. 4b).

For Src peptides, both Nt-acetylated and Nt-free variants pulled down DCAF10 with the Nt-free peptide retaining 62.0 ± 8.7% of the Nt-acetylated signal, consistent with the AF3-predicted weaker pocket engagement (Fig. 2o and Supplementary Fig. 4b). Although our initial peptide pull-downs showed significant binding of DCAF10 to Ac-Src (-log₂ 5; Fig. 1c) with few additional interactors recovered (Fig. 1c), the reduced selectivity suggests that additional cellular factors may enhance Ac-Src–DCAF10 engagement in cells. Direct comparison of Nt-acetylated Lyn, Fyn, and Src showed a rank order of Lyn (Lyn ≅ 100%) ≈ Fyn (94 ± 6%) > Src (70 ± 16%) (Supplementary Fig. 4c, d). In contrast, neither Nt-acetylated nor Nt-free THOC7 peptides or GST alone bound DCAF10 under these conditions (Fig. 2p).

Collectively, DCAF10 directly engages acetylated N-termini of typically myristoylated SFKs, with Src representing a poorer fit to the pocket and acidic flanks reducing affinity.

## SFKs are NatA substrates in vitro

Gly-starting proteins can be substrates of the Nt-acetyltransferase NatA[3,21]. Having established direct DCAF10 binding to Ac-Gly N-termini, we next asked whether SFK N-termini are bona fide NatA substrates. We developed an in vitro Nt-acetylation assay comprising purified NatA, acetyl-CoA (Ac-CoA), and the peptide of interest (Fig. 3a, b)[33]. Acetyl-group transfer was assessed using CPM [7-(diethylamino)−3-(4′-maleimidophenyl)−4-methylcoumarin], which reacts specifically with free thiol groups to produce a highly fluorescent product (excitation: 390 nm; emission: 479 nm), allowing detection of CoA production after acetyl-group transfer to the peptide (Supplementary Fig. 4e). We optimized the conditions to determine Michaelis–Menten kinetics for NatA-mediated Nt-acetylation with a target peptide (SASEAGVRWG) previously used for NatA assays and co-crystallization (Supplementary Fig. 4f)[34]. The peptide exhibited a $K_m$ of 8.4 ± 1.7 µM. Under the same conditions, Src was also Nt-acetylated but exhibited a -17-fold higher $K_m$ value (143.8 ± 24 µM) (Fig. 3c). Negative controls, including a Pro-starting peptide from COX17 (Uniprot ID: Q14061; Pro-starting proteins are typically not acetylated[24]) and pre-acetylated Src, showed no detectable acetylation. Lyn and Fyn did not show Nt-acetylation under these conditions. Likely, this was due to a $K_m$ even higher than that of Src, and implying that the peptide concentrations in the assay might have been insufficient for the reaction to proceed with an appreciable rate. In line with this expectation, increasing the starting Lyn and Fyn peptide concentrations to 1 mM (from 160 µM in the original assay) revealed detectable Nt-acetylation, with a calculated $K_m$ of 411.1 ± 94 µM for Lyn and 235.1 ± 46 µM for Fyn (Fig. 3d and Supplementary Fig. 4g). Negative controls (COX17 and Ac-Lyn) produced low

signals. In conclusion, Src, Lyn and Fyn are poor NatA substrates with high $K_m$ values > 100 µM. Given the direct association of NatA with SFK N-termini at the ribosomal exit tunnel, however, we surmise that Nt-acetylation of SFKs in cells may proceed despite these unfavorable kinetic parameters.

## DCAF10 regulates SFK levels upon NMT1/2 depletion

Our biochemical and modeling data indicate that DCAF10 directly engages acetylated N-termini of typically myristoylated SFKs (with acidic flanks weakening binding). This supports a working model in which, when myristoylation fails, an Ac-Gly is generated for the CUL4A-DDB1-DCAF10 E3 ligase, wherein DCAF10 acts as substrate-binding receptor. Consistent with co-translational action of NMT and NatA on overlapping Nt-substrates, lowering NMT1/2 is expected to bias nascent SFKs toward mutually exclusive fates—Nt-acetylated or Nt-free—routing them either to CUL4A-DDB1-DCAF10 or to the published CUL2-elonginB/C-ZYG11B/ZER1 Gly/N-degron pathway, respectively[14,21,35].

We therefore asked whether DCAF10 contributes to changes in SFK steady-state levels upon NMT1/2 depletion and treated HeLa cells with siRNAs targeting NMT1/2, DCAF10, or both (Fig. 4a). In NMT1/2-depleted cells, Lyn levels were strongly reduced compared to untreated cells (53.3 ± 20.7%), Fyn showed a slight reduction (12.03 ± 2.5%), and Src showed no reduction (3.3 ± 4.6%) (Fig. 4b, c–e, left panels). Despite these baseline differences, Lyn, Fyn, and Src levels increased after DCAF10 depletion with a stronger effect when NMT1/2 was co-depleted with DCAF10 (Lyn increase: 1.78 ± 0.75-fold versus 3.3 ± 1.5-fold; Fyn increase: 1.73 ± 1-fold versus 4.3 ± 2.3-fold; Src increase: 1.1 ± 0.3-fold versus 1.7 ± 0.5-fold; Fig. 4b, c–e right panels), consistent with DCAF10 contributing to SFK degradation following loss of NMT1/2. THOC7 levels slightly increased after NMT1/2 siRNA (12.13 ± 32.5%), and DCAF10 siRNA did not change THOC7 levels upon NMT1/2 depletion (1.2 ± 0.4-fold versus 1.2 ± 0.6-fold; Fig. 4b, f). DCAF10 was not detectable using multiple anti-DCAF10 antibodies. To overcome this limitation, we employed high-pH fractionation of the whole proteome combined with MS, which confirmed a greater than 256-fold reduction in DCAF10 levels following siRNA treatment (Fig. 4b and Supplementary Data 4).

In DLD-1 (colorectal cancer) and RPE-1 (non-transformed, near-diploid retinal pigment epithelium), co-depletion of DCAF10 with NMT1/2 increased Lyn, Fyn, and Src relative to NMT1/2 knockdown (KD) alone (Fig. 4g, h). Conversely, FLAG-DCAF10 overexpression in the context of NMT1/2 KD further decreased Lyn (-2.1-fold compared to control cells), while THOC7 remained unchanged (Supplementary Fig. 5a–c), supporting a role for DCAF10 in modulating SFK levels under reduced NMT1/2.

While these measurements were directed at determining the steady-state concentrations of our proteins of interest, we reasoned that if loss of myristoylation increases SFK degradation, accelerated turnover should be detectable even when steady-state abundance does not decline. To assess this, HeLa cells were transfected with NMT1/2 siRNA or left untreated and analysed after addition of cycloheximide (8 h) to block protein synthesis or MG132 (4 h) to inhibit

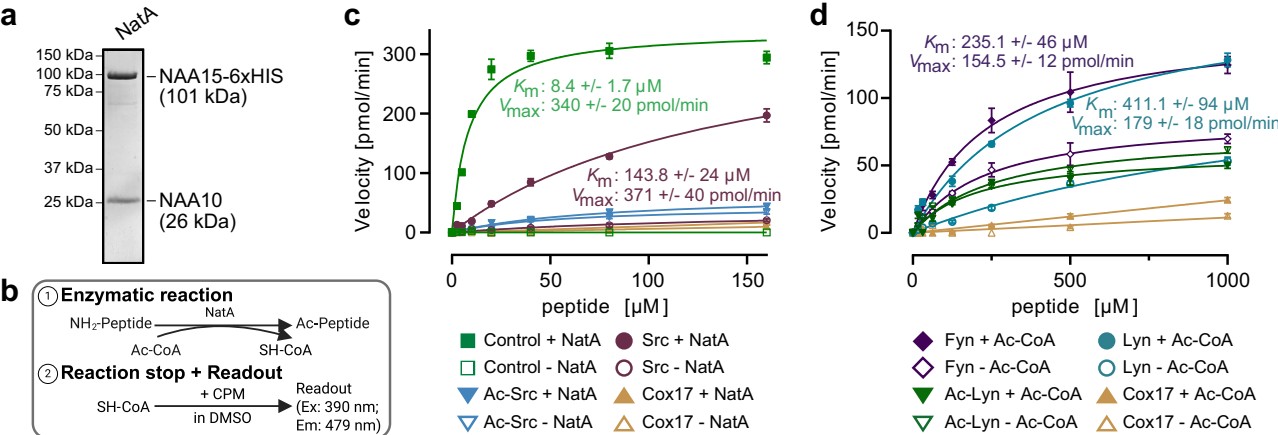

**Fig. 3 | Src-family kinases are NatA targets in vitro. a** Purification of human NatA, consisting of NAA15 and NAA10. Coomassie staining is shown. **b** Schematic of in vitro Nt-acetylation assay (adopted from[33]). Created in BioRender. Bange, T. (2025) https://BioRender.com/fl0ijrm. **c, d** Michaelis–Menten kinetics of in vitro Nt-acetylation assays. Assays contained 20 nM NatA, 80 μM acetyl-CoA (Ac-CoA), and a dilution series of Nt-peptides. The reaction was stopped after 60 min (**c**) or 120 min

(**d**), and Nt-Ac detected by CPM fluorescence (excitation 390 nm, emission 479 nm). Reaction velocities (pmol/min) are plotted against peptide concentrations; $K_m$ (Michaelis–Menten) and maximum velocity ($V_{max}$) were determined using GraphPad Prism. Data represent the mean ± standard deviation (s.d) ($n = 4$ independent enzyme assays).

proteasomal degradation. Cycloheximide further enhanced the reduction of Lyn, Fyn, and Src in NMT1/2 KD cells relative to cycloheximide-treated control cells (Lyn decrease: 2-fold; Fyn decrease: 1.5-fold; Src decrease: 1.3-fold), whereas MG132 caused accumulation (Lyn increase: 1.6-fold; Fyn: 1.6; Src: 2-fold) (Supplementary Fig. 5d–g). THOC7 levels remained unaffected by NMT1/2 KD under all conditions (THOC7 decrease: 1.1-fold; THOC7 increase: 1.1-fold) (Supplementary Fig. 5d, h). Together, these cycloheximide and MG132 experiments show that NMT1/2 KD accelerates Lyn, Fyn, and Src degradation; for Fyn and Src, this occurs despite modest or no steady-state changes, likely because faster degradation is partially balanced by accelerated ongoing synthesis.

In summary, our data indicate that reduced N-myristoylation accelerates SFK turnover, with DCAF10 playing a critical role in the enhanced degradation of Lyn, Fyn, and Src. Additionally, these findings suggest a translation-degradation feedback loop to buffer steady-state levels even as degradation rates increase at least for Fyn and Src.

## CUL2-elonginB/C-ZYG11B/ZER1 and CUL4A-DDB1-DCAF10 act complementarily after NMT1/2 knockdown

Having established DCAF10-dependent degradation upon loss of N-myristoylation, we next asked whether DCAF10 operates in parallel with the Gly/N-degron branch. We therefore performed combinatorial siRNA depletions of NMT1/2 with DCAF10 and/or ZYG11B/ZER1 (Fig. 4i). Consistent with the preceding data, DCAF10 depletion together with NMT1/2 KD increased Lyn, Fyn, and Src levels (Lyn: 3.2-fold; Fyn: 3.5-fold; Src: 1.4-fold). ZYG11B/ZER1 depletion increased Lyn and Src relative to NMT1/2 KD alone, but had no effect on Fyn (Lyn: 2.3; Fyn: 1; Src: 1.7-fold). However, simultaneous depletion of DCAF10 and ZYG11B/ZER1 together with NMT1/2 produced the strongest stabilization: Lyn 4.9-fold (Fig. 4j, k); Fyn 5.3-fold (Fig. 4j, l); Src 2.2-fold (Fig. 4j, m). As with DCAF10, we did not succeed detecting endogenous ZYG11B using commercial antibodies; however, high-pH fractionation of the whole proteome, combined with MS, confirmed an at least 16-fold reduction in ZYG11B levels (Fig. 4j and Supplementary Data 5).

Together, these data support complementary surveillance by both the DCAF10 and ZYG11B/ZER1 branches on SFKs, with relative contributions differing among family members: greater DCAF10 sensitivity for Lyn and Fyn, and of ZYG11B/ZER1 for Src, respectively.

## CUL4A-DDB1-DCAF10 ubiquitinates SFKs in vitro

Having established cellular regulation by DCAF10 under reduced N-myristoylation, we aimed to test whether CUL4A-DDB1-DCAF10 directly ubiquitinates SFKs. To investigate this, we reconstituted a catalytically active E3 ligase complex (WT) comprising full-length CUL4A, DDB1, DCAF10, and the catalytic subunit RBX1. In addition, we generated a catalytically inactive variant (RBX1[mut]; C75A, H77A) and a control complex lacking DCAF10 (-DCAF10) (Fig. 5a, b and Supplementary Fig. 6a–c, e–g, i–k). Lyn, Fyn, and Src were immunoprecipitated from lysates of cells either depleted of NMT1/2 or left untreated (Supplementary Fig. 5i) and subsequently subjected to in vitro ubiquitination assays using the three purified ligase complexes (WT, RBX1[mut], and -DCAF10) (Fig. 5c, e, g). Ubiquitination signals were quantified as ratios relative to RBX1[mut]. For Lyn, Fyn, and Src, the WT/RBX1[mut] ratio was >2-fold in immunoprecipitates (IPs) from NMT1/2-depleted cells (Lyn: 2.18; Fyn: 2.27; Src: 2.03) and >2.5-fold in IPs from control cells (Lyn: 2.58; Fyn: 2.73; Src: 3.0). Conversely, the -DCAF10/RBX1[mut] ratio was 1 under both conditions (Fig. 5d, f, h). As we expect NMT1/2 depletion to cause a destabilization of SFKs, their equal or even greater ubiquitination in IPs from untreated cells relative to NMT1/2-depleted cells was unanticipated. We speculate that (i) Nt-acetylated SFKs may degrade rapidly, preventing their accumulation upon NMT1/2 depletion; (ii) an acetylated fraction of SFKs may exist under physiological conditions; (iii) DCAF10 might recognize additional surfaces. Nonetheless, these data collectively demonstrate DCAF10-dependent ubiquitination of SFKs by CUL4A-DDB1-DCAF10 in vitro.

To directly test the contribution of the predicted Ac-Gly binding pocket, we purified a DCAF10 triple mutant (F172G, K257A, and I475G) that, based on AF3 models, is expected to disrupt pocket integrity and consequently SFK binding (Fig. 5i, j and Supplementary Fig. 6d, h, l). In vitro ubiquitination assays showed that the resulting mutant abolished ubiquitination of Lyn and Fyn, whereas Src ubiquitination was unaffected by the mutations, while remaining DCAF10-dependent (Fig. 5k). This pattern, together with AF3 models and streptavidin pull-downs (e.g., Src in Fig. 2k, l, o versus Lyn in Fig. 2c, e, m), suggests that the Nt region of Src is recognized by the same DCAF10 pocket/surface, but engages it more shallowly. Thus, Lyn and Fyn require an intact Ac-Gly pocket for ubiquitination, whereas Src tolerates pocket-weakening substitutions.

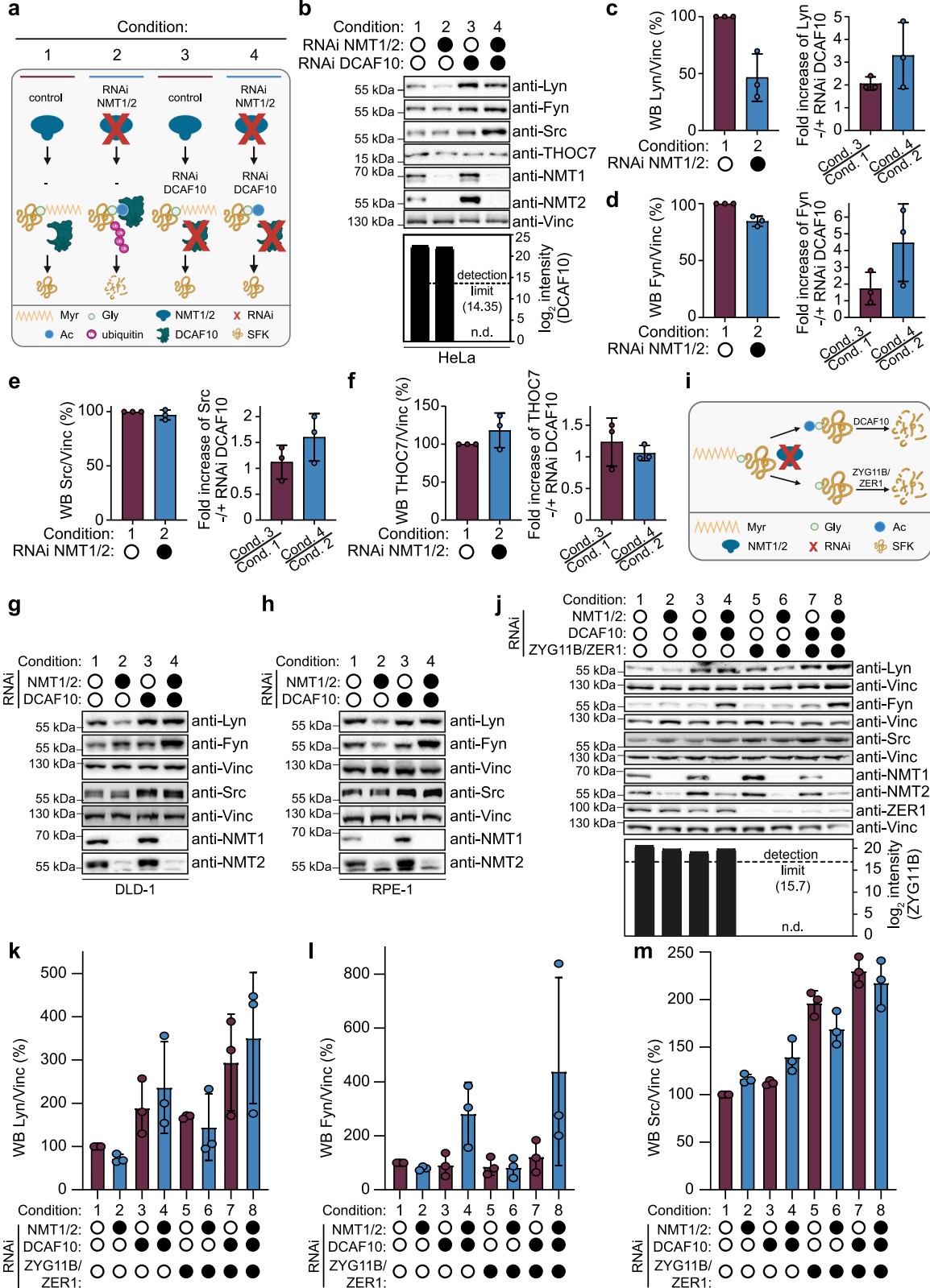

## N-terminal glycine mutation controls DCAF10-dependent ubiquitination and full-length interaction

To further investigate the role of the Nt-amino acid and its modification in DCAF10 recognition and DCAF10-dependent ubiquitination, we used the CRISPR/Cas9 technique to knock out Lyn in DLD-1 cell lines with T-REx Flp-In background (DLD-1 Lyn KO Flp-In T-REx) (Supplementary Fig. 7a–c). We then integrated Lyn[WT]-GFP and two Nt-mutants

(Lyn[G2A]-GFP and Lyn[G2P]-GFP) under a doxycycline-inducible promoter into the Flp-In cassette. Alanine (G2A) is expected to be efficiently acetylated by NatA, generating an Nt-acetylated Lyn mutant. Conversely, the proline (G2P) mutant, is expected not to be acetylated and may retain a free N-terminus[3,36,37]. Doxycycline-induced expression of Lyn[WT]-GFP, Lyn[G2A]-GFP, and Lyn[G2P]-GFP was confirmed by Western blot, MS, and fluorescence microscopy (Supplementary Figs. 7d, e; 8a and

**Fig. 4 | DCAF10 regulates Src-family kinase levels upon NMT1/2 depletion and acts complementarily to ZYG11B/ZER1. a** Schematic of experiments. Created in BioRender. Bange, T. (2025) https://BioRender.com/ute0t5z (**b–h**). **b** HeLa cells were treated for 72 h with different siRNA combinations as indicated. Representative WBs are shown. DCAF10 levels from mass spectrometry (MS) are shown as log2 intensities (bar plot). Dashed line = MS detection limit. Filled circles indicate siRNA treatment, empty circles indicate untreated. N.d. not detected. Left: WB quantifications for **c** anti-Lyn, **d** anti-Fyn, **e** anti-Src, and **f** anti-THOC7 with/without NMT1/2 siRNA normalized to vinculin (Vinc). Right: bar plots report the effect of **c** Lyn, **d** Fyn, **e** Src and **f** THOC7 levels of DCAF10 knockdown as a fold-change ratio relative to the matched control in each background; purple: DCAF10 siRNA)/untreated control; blue: (NMT1/2 + DCAF10 siRNA)/NMT1/2 siRNA alone. WB signals were normalized to vinculin. Mean ± s.d. with individual data points are shown (*n* = 3 independent biological replicates). **g**, **h** Same experiments as (**b**) using **g** DLD-1 (colorectal adenocarcinoma cell line) and **h** RPE-1 (hTERT-immortalized retinal pigment epithelial cells). **i** Schematic of possible SFK N-termini in experiments. Created in BioRender. Bange, T. (2025) https://BioRender.com/efmhrz6 (**j–m**). **j** HeLa cells were treated for 72 h with different siRNA combinations of NMT1/2, DCAF10, and ZYG11B/ZER1. Representative WBs and MS quantification of ZYG11B are shown. Treatments are indicated as filled circles, untreated by empty circles. WB quantifications of **k** anti-Lyn, **l** anti-Fyn and **m** anti-Src. Mean ± s.d. with individual data points are shown (*n* = 3 independent biological replicates). Source data are provided as a Source data file.

Supplementary Data 6). While all constructs were ~2–5-fold higher expressed than endogenous Lyn, G2A and G2P were further elevated relative to WT, which may reflect increased stability when Nt-Gly/acetylation is altered (Supplementary Fig. 7e). Notably, Lyn$^{WT}$-GFP strongly localized to the plasma membrane, as expected, whereas both mutants exhibited diffuse cytoplasmic staining, indicating that correct modification of the Nt-residue (myristoylation) determines membrane localization (Supplementary Fig. 8b)[27,28]. Lyn$^{WT}$-GFP mirrored the behavior of endogenous Lyn under NMT1/2 siRNA treatment. Lyn levels decreased with NMT1/2 KD, but were rescued when siRNAs targeting both NMT1/2 and DCAF10 were applied (Fig. 6a, b). Specifically, endogenous Lyn levels decreased by 31% on average following NMT1/2 KD confirming previous results, whereas Lyn$^{WT}$-GFP levels decreased by 21% on average. Upon combined KD of NMT1/2 and DCAF10, endogenous Lyn levels increased by 36.5%, while Lyn$^{WT}$-GFP levels rose by 41.7% (Fig. 6b). The weaker reduction of Lyn$^{WT}$-GFP may result from Lyn overexpression and the limiting availability of DCAF10. Nevertheless, Lyn$^{WT}$-GFP levels remained dependent on NMT1/2 and DCAF10, whereas Lyn$^{G2A}$-GFP and Lyn$^{G2P}$-GFP did not show such dependency, demonstrating that acetylation and the presence of glycine are both required for DCAF10-dependent destabilization (Fig. 6a, b). Lyn$^{G2A}$-GFP showed a slight increase after DCAF10 depletion, independently of NMT1/2, possibly reflecting a residual interaction with acetylated alanine. Confirming this, peptide pull-downs comparing Ac-Gly (Ac-Lyn$^{WT}$), Ac-Ala (Ac-Lyn$^{G2A}$), and Nt-free Pro (Lyn$^{G2P}$) peptides revealed that the Ac-Ala peptide interacted with DCAF10, but at 15-fold lower levels relative to the Ac-Gly peptide and was only identified in 2 of 3 replicates (Supplementary Fig. 7f and Supplementary Data 7). Conversely, Nt-free Pro peptides failed to bind DCAF10 in all replicates, in line with the cellular experiments (Supplementary Fig. 7g and Supplementary Data 7). Together, these results support the requirement of Nt-acetylation and glycine for DCAF10 binding.

We next assessed the role of the Lyn Nt-residue for ubiquitination by performing two complementary experimental set-ups (Fig. 6c). First, we performed Lyn$^{WT}$-GFP IPs from cells either treated with NMT1/2 siRNA or left untreated and directly blotted for ubiquitin (Fig. 6d–f). Lyn$^{WT}$-GFP was ubiquitinated under both conditions, with a stronger signal in controls (as above), whereas G2A and G2P showed no detectable ubiquitination.

Next, we tested DCAF10-dependent in vitro ubiquitination of Lyn$^{WT}$-GFP IPs with the previously generated three DCAF10 complexes (WT, RBX1$^{mut}$, -DCAF10) under NMT1/2 siRNA and control conditions (Fig. 6d, g). Lyn$^{WT}$-GFP was ubiquitinated by the WT E3 complex in both conditions; however, unlike in vitro ubiquitination with endogenous Lyn (Fig. 5c, d), ubiquitination increased ~6-fold under NMT1/2 siRNA conditions. This is consistent with DCAF10 being limiting and overexpressed Lyn-GFP, accumulating an acetylated fraction available for DCAF10 in vitro. Then, we performed IPs for all Nt variants under NMT1/2 siRNA conditions (Fig. 6d, h–i). Lyn$^{WT}$-GFP was robustly ubiquitinated by the WT E3 complex. In contrast, Lyn$^{G2A}$-GFP and Lyn$^{G2P}$-GFP ubiquitination by the WT complex was strongly reduced (Lyn$^{G2A}$-GFP: -82 ± 13%; Lyn$^{G2P}$-GFP: -84 ± 10%), and RBX1$^{mut}$ and -DCAF10

complexes failed to ubiquitinate any variant above background (Fig. 6h, i). In vitro ubiquitination assays using untreated cell lysates led to similar results (Supplementary Fig. 7h, i). These results are consistent with the idea that Ac-Gly is the primary driver of DCAF10-dependent ubiquitination.

Finally, we asked whether Lyn$^{WT}$-GFP interacts with DCAF10 in the full-length context. To this end, we performed anti-GFP IPs from doxycycline-induced or uninduced Lyn$^{WT}$-GFP cell lines and analysed binding partners by MS. Lyn$^{WT}$-GFP significantly interacted with DCAF10 under both NMT1/2 siRNA and control conditions, with significant enrichment upon NMT1/2 KD (Fig. 6j–l and Supplementary Data 8), confirming interactions of the full-length proteins. Under control conditions, interactors were enriched for cell/plasma membrane, lipoprotein, and GPI-anchor categories (GO-terms), consistent with correct membrane localization; under NMT1/2 KD, binding partners in the ribosomal and nucleolus categories were enriched, consistent with delocalization (Supplementary Fig. 7j, k and Supplementary Data 8).

Collectively, our data confirm Ac-Gly as the major determinant for DCAF10-dependent ubiquitination and full-length interaction.

## Discussion

N-degron pathways detect omitted co-translational modifications and can target proteins with altered N-termini for ubiquitination and subsequent degradation. When N-myristoylation is reduced, nascent SFKs can present two mutually exclusive N-termini. Nt-free glycine is recognized by the established CUL2-elonginB/C-ZYG11B/ZER1 branch[14]. Convergent biochemical, genetic, and reconstitution evidence presented supports that Nt-acetylated glycine engages DCAF10, prompting CUL4A-DDB1-DCAF10-dependent ubiquitination and degradation. Properly modified SFKs are degraded via internal degrons, e.g., by CBL (Casitas B-lineage lymphoma protein)-dependent routes under defined stimuli[38]. Although our mechanistic evidence focuses on SFKs, DCAF10 preferentially binds Nt-acetylated N-termini from several typically myristoylated proteins, suggesting broader relevance that remains to be established. Pending proteome-wide in vivo validation, we provisionally refer to this DCAF10 branch as the Ac-Gly/MO (myristoylation-omitted) N-degron pathway (Fig. 7a, b).

We demonstrate that both Nt-acetylation and Gly2 are required for DCAF10 recognition. THOC7, although acetylated at Gly2, is insensitive to DCAF10 depletion, indicating additional sequence constraints. AF3 predictions suggest that residues 3–4 contribute to pocket engagement. Notably, in silico substitution of Asn4 with Ile in Src is predicted to position the acetyl group deeply within the DCAF10 pocket, resembling Lyn and Fyn, consistent with a preference for hydrophobic residue(s) near position 4 (Supplementary Fig. 9). Our swap experiments show that acidic residues at positions 5–7 impede binding, whereas neutralization (Ala-Ala-Ala at position 5–7) preserves it, revealing a downstream charge filter. Proteome-scale "glyome" analyses report that Ser at position 6 and Lys at position 7 are among the most conserved features of N-myristoylated Gly-starting N-termini, whereas Nt-acetylated counterparts are enriched for acidic residues at

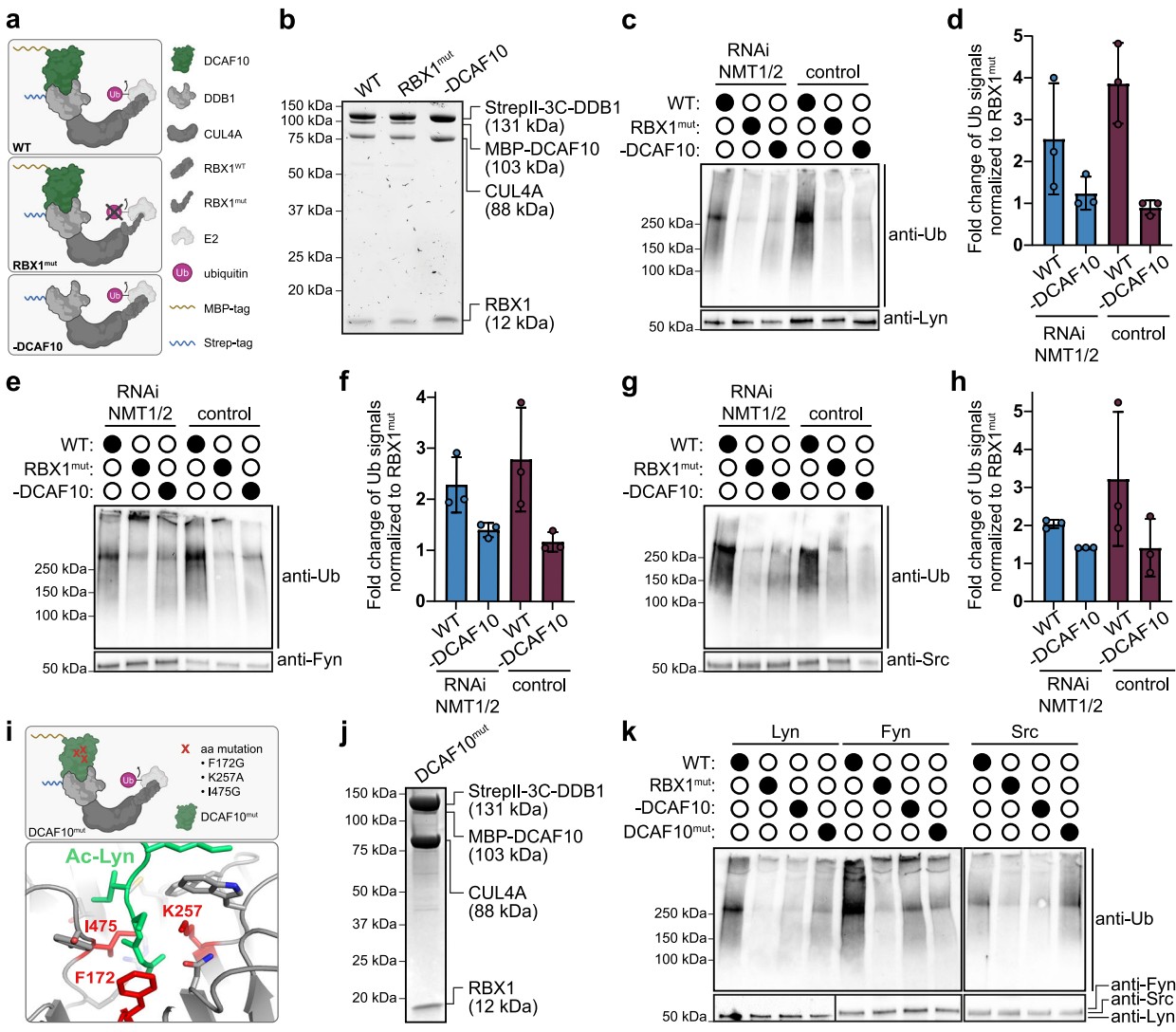

**Fig. 5 | CUL4A-DDB1-DCAF10 ubiquitinates Src-family kinases in vitro.**
**a** Schematic of the three purified E3 ligase complexes together with E2 and ubiquitin; wild-type (WT) complex: CUL4A, RBX1, MBP-DCAF10, and StrepII-DDB1; mutant (RBX1mut) complex: WT with RBX1C75A/H77A disrupting the catalytic activity; -DCAF10 complex: WT components without DCAF10. Created in BioRender. Bange, T. (2025) https://BioRender.com/jwsclv4. **b** TCE (2, 2, 2-trichloroethanol) in-gel visualization of WT, mut, and -DCAF10 complexes. In vitro ubiquitination of **c** Lyn, **e** Fyn and **g** Src immunoprecipitates with WT, mut, and -DCAF10 complexes. HeLa lysates were treated with or without NMT1/2 siRNA (72 h). Representative WBs are shown. **d, f, h** WB quantifications of in vitro ubiquitinations of **c, e, g** showing fold change as ratios relative to RBX1mut: WT/RBX1mut and -DCAF10/RBX1mut. Ub signals were normalized to the bait: **d** Lyn, **f** Fyn and **h** Src. Blue: NMT1/2 siRNA, purple: control. Mean ± s.d. with individual data points are shown (n = 3 independent biological replicates). **i** Schematic of DCAF10mut and AlphaFold 3 prediction of DCAF10 (gray) and Ac-Lyn (green); mutated residues highlighted in red. Created in BioRender. Müller, F. (2025) https://BioRender.com/sq7z7oz. **j** TCE (2,2,2-Trichloroethanol) in-gel visualization of DCAF10mut complex. **k** In vitro ubiquitination of Lyn, Fyn, and Src immunoprecipitates with WT, RBX1mut, -DCAF10 and DCAF10mut complexes. Representative WBs are shown. Source data are provided as a Source data file.

positions 6–7[21,39]. Notably, positions 6 and 7 of DCAF10 binding peptides often carry Ser and Lys, providing a local positive charge typical of N-myristoylation motifs. Thus, DCAF10 recognition aligns with a non-acidic, positively biased subset—Ac-Gly combined with the absence of acidic residues at 5–7—indicating surveillance of a defined subset of acetylated Gly N-termini beyond SFKs.

DCAF10 is insufficiently studied, yet its high expression correlates with better ovarian cancer survival, while its loss predicts poor lung adenocarcinoma prognosis, suggesting a protective role[40,41]. It stabilizes pro-apoptotic OTUD1, promoting MCL1 degradation and apoptosis[42]. DCAF10 also targets the adenoviral protein E1A, likely via a phospho-dependent degron, indicating recognition of distinct degrons[30]. Proximity-labeling (BioID) data identify several DCAF10-associated proteins that are typically N-myristoylated, including PPM1G, PPM1B, ARL3, DDX46, FAM49B, and Yes/Src, suggesting a

potential intersection between DCAF10 interactors and N-myristoylated proteins[43]. Importantly, our binding and modeling data suggest that DCAF10 recognizes N-terminally Ac-Gly rather than Myr-Gly per se.

DCAF10 depletion increases Lyn, Fyn, and Src levels modestly even without NMT1/2 KD, and DCAF10-dependent SFK ubiquitination is detectable under basal conditions as well as upon NMT loss, consistent with a physiological acetylated fraction of SFKs. Proteome-scale mapping and motif analyses indicate overlap between N-myristoylation and Nt-acetylation targets, supporting the existence of alternatively processed N-termini in vivo. In addition, ARF1 is frequently observed as mixed myristoylated/non-myristoylated pools, illustrating incomplete modification[21,26]. These observations align with the general role of N-degron pathways in protein quality control—removing species that fail to acquire a membrane anchor—but also

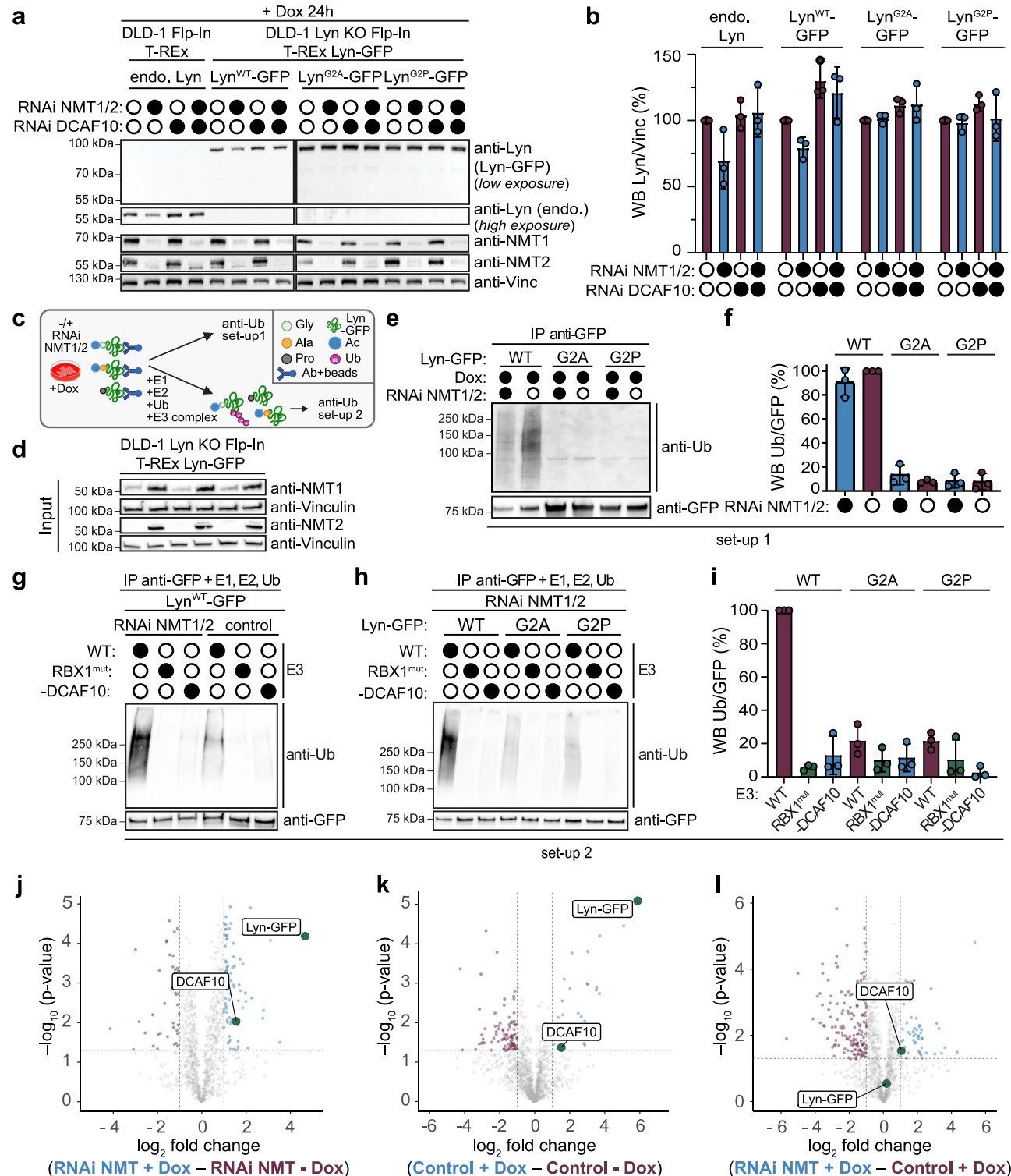

raise the possibility of a regulatory function, whereby readout of co-translational Nt-states influences steady-state kinase abundance and localization. Moreover, the branch taken is not uniform: stabilization patterns and modeling indicate that Lyn and Fyn are preferentially surveyed by DCAF10, whereas Src relies more on ZYG11B/ZER1—suggesting that sequence context around Gly2 (positions 3–7) can bias routing and thereby fine-tune SFK abundance and localization.

N-myristoylation anchors SFKs at membranes. When it is reduced (e.g., G2A/C3A mutants), SFKs can accumulate in the nucleus upon export inhibition, indicating that myristoylation normally limits

nuclear retention[22,44–47]. Our data suggest that non-myristoylated SFKs become Nt-acetylated, creating recognition-competent N-termini for DCAF10 surveillance and pointing to compartmentalized control of abundance and signaling. The balance between degradation and nuclear residence may be influenced by the co-translational fraction of Ac-starting vs. Nt-free species (NatA vs. NMT), the cell-state–dependent availability/specificity of Ac/N-recognins (e.g., DCAF10 vs. ZYG11B/ZER1) together with sequence context at positions 3–7, and the exposure of basic motifs/NLS-like elements versus nuclear export. Given that nuclear SFKs have context-dependent oncogenic associations, the Ac-Gly/MO pathway may fine-tune localization and

**Fig. 6 | N-terminal glycine mutation abolishes DCAF10-dependent stability and ubiquitination. a** DLD-1 Flp-In T-REx cells and Lyn KO cells expressing Lyn^WT-GFP, Lyn^G2A-GFP, or Lyn^G2P-GFP were induced for 24 h with 100 ng/mL doxycycline. Cells were treated with NMT1/2 siRNA, DCAF10 siRNA, or both (or left untreated) for 72 h. Representative WBs are shown. Filled circles represent siRNA treatment, empty circles untreated. **b** WB quantifications of Lyn levels normalized to vinculin. Purple: untreated with NMT1/2 siRNA; blue: treated with NMT1/2 siRNA. Mean ± s.d. with individual data points are shown (n = 3 independent biological replicates). **c** Schematic of ubiquitination experiments. Created in BioRender. Bange, T. (2025) https://BioRender.com/920mc12 (**e–i**). **d** WBs of lysates used for direct and in vitro ubiquitination (**e–i**). **e** N-terminal Lyn-GFP variants were immunoprecipitated using anti-GFP beads and blotted against ubiquitin. WBs against Ub and GFP are shown. **f** Quantification of (**e**); mean ± s.d. with individual data points are shown (n = 3 independent biological replicates) **g** Lyn^WT-GFP was immunoprecipitated using

anti-GFP beads and then subjected to in vitro ubiquitination assays with the DCAF10 WT, RBX1^mut and -DCAF10 complexes in the presence or absence of NMT1/2 siRNA treatment. **h** N-terminal Lyn-GFP variants were immunoprecipitated using anti-GFP beads and subjected to in vitro ubiquitination assays with WT, RBX1^mut, and -DCAF10 complexes. Lysates have been treated with NMT1/2 RNAi for 72 h. **i** Quantification of in vitro ubiquitination assays. Mean ± s.d. with individual data points are shown (n = 3 independent ubiquitination assays). Volcano plots of Lyn-GFP^WT binding partners: **j** NMT1/2 siRNA ± doxycycline (Dox); **k** Control ± doxycycline; **l** NMT1/2 siRNA vs. control both with doxycycline. The −log10 adjusted p-value (two-sided Student's t test with permutation-based multiple-testing correction; y-axis) is plotted against the log2 fold change (x-axis). Threshold for significance: −log10 p-value ≥ 1.3 (p ≤ 0.05), log2 fold change ≤ −1 or ≥ 1. DCAF10 and Lyn-GFP are marked in green, significant binding partners in purple and in blue, respectively. Source data are provided as a Source data file.

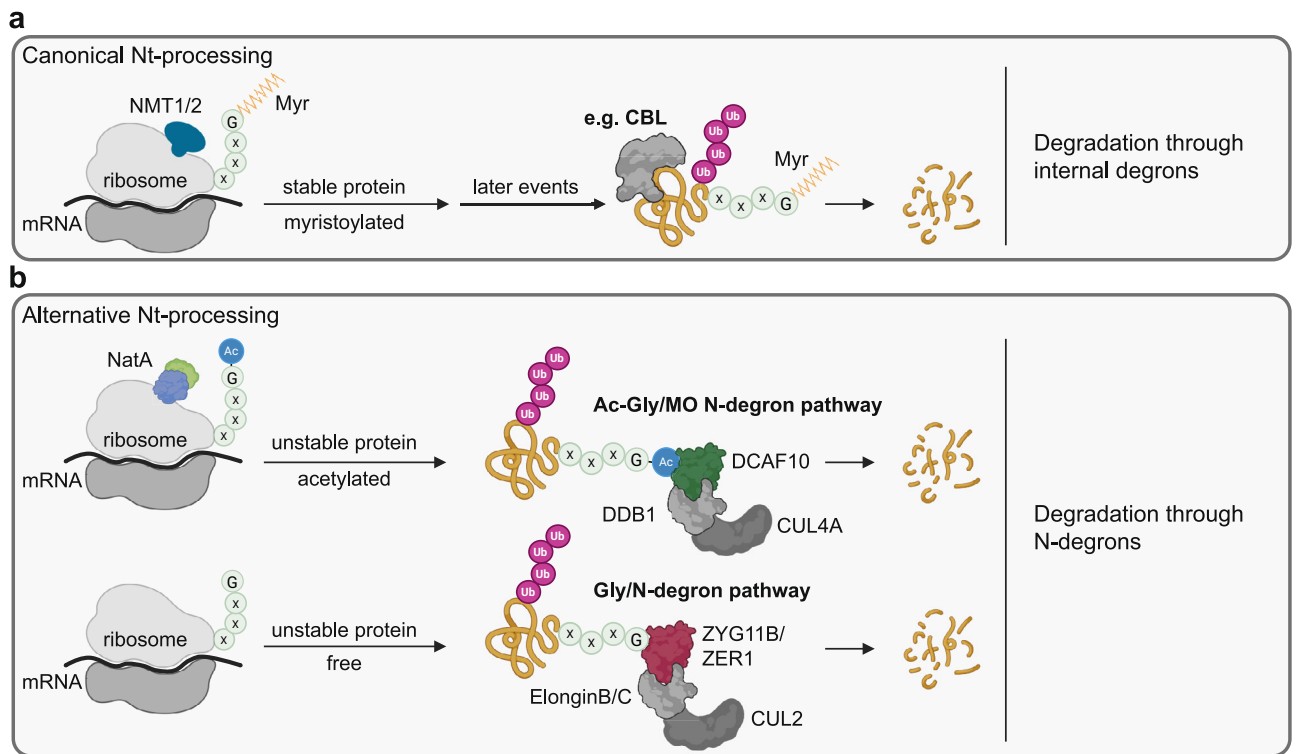

**Fig. 7 | Scheme of degradation pathways for Gly-starting proteins. a** Gly-starting proteins correctly modified by N-myristoylation (Myr) are stable and degraded via internal degrons when required for function. **b** Gly-starting proteins with alternative Nt-processing omitting N-myristoylation. Free glycine residues are recognized by CUL2-elonginB/C-ZYG11B/ZER1 (Gly/N-degron pathway), whereas proteins

undergoing Nt-acetylation instead of N-myristoylation are targeted by the CUL4A-DDB1-DCAF10 (Ac-Gly/MO N-degron pathway). CBL Casitas B-lineage lymphoma protein; NMT N-myristoyltransferase 1 and 2; NatA Nt-acetyltransferase A. Created in BioRender. Bange, T. (2025) https://BioRender.com/tn5ptiw.

output in vivo[48,49]. Therapeutically, partial NMT inhibition could delocalize SFKs from membranes while exposing Ac-Gly or Gly/N-degrons to promote turnover; net outcomes will depend on kinase identity, sequence context around Gly2 (positions 3–7), and the relative activities of the DCAF10 and ZYG11B/ZER1 branches.

## Methods

All peptides, cell lines, siRNAs and oligonucleotides, plasmids and primers, antibodies and reagents used in this study are listed in Supplementary Tables 1–10. Detailed protocols for protein purification, peptide pull-downs, and common procedures (siRNA transfection, fluorescence microscopy, immunoprecipitation) are provided in the Supplementary Methods. Source data are provided as a Source data file with all raw Western blots (both those shown in the manuscript and

those used for quantification), together with the corresponding quantification data.

## Peptide pull-downs

Biotinylated 10-mer Nt-peptides (Nt-free, Nt-acetylated, and, where indicated, Nt-myristoylated; sequences in Supplementary Table 1) were synthesized with a C-terminal Lys₃ spacer and biotin, immobilized on magnetic streptavidin beads, and incubated with HeLa or mouse liver lysates. After standardized washes, bound proteins were on-bead reduced/alkylated, digested with LysC/trypsin, desalted, and analysed by LC-MS in DIA mode (Orbitrap Exploris 480) as described previously[24]. Variants included endogenous Nt-sequences, swap and mutant (Ala-Ala-Ala or Nt-amino acids) peptides, and pairwise comparisons of conditions of interest. Full buffer compositions, bead

loading, wash schemes, and MS parameters are provided in the Supplementary Methods.

### AF3 prediction of DCAF10 and acetylated SFKs

All DCAF10-peptide structure predictions were done with AlphaFold 3.0.0 (AF3)[32]. The Nt-acetylation was defined using the "alphafold3" dialect with the ccdCodes ligand "ACE" bonded to the nitrogen atom of the first glycine residue in the peptide. The fragment DCAF10 aa 120–559 that lacks the 119 N-terminally disordered residues was used for all predictions (UniprotID: Q5QP82). The long disordered loop (residues 347–393) was included. Conservation of DCAF10 was calculated as previously described[50].

### In vitro Nt-acetylation assay

The assay was performed essentially as described previously[33]. Briefly, peptide substrates (Supplementary Table 1) were incubated with purified human NatA and acetyl-CoA; acetylation was quantified by CPM fluorescence (CoA detection), and Michaelis–Menten parameters were derived from 8-point 1:1 dilution series within the linear range ($n = 4$ independent enzymatic assays). Full buffer compositions, reagent concentrations, plate format, instrument settings, controls, and fitting procedures are provided in the Supplementary Methods and Tables.

### In vitro ubiquitination assay

In vitro ubiquitination of endogenous Lyn, Fyn, Src, and Lyn-GFP variants was carried out as described[11] with a reconstituted CUL4A-DDB1-DCAF10-RBX1 E3 ligase (WT), a catalytically inactive RBX1 mutant (RBX1$^{mut}$; C75A, H77A), a −DCAF10 control, and a pocket DCAF10 mutant (DCAF10$^{mut}$; F172G, K257A, I475G). Reactions contained E1, UBCH5B (E2), ubiquitin, Mg-ATP in ligation buffer and were incubated with bead-bound substrates, followed by washing and immunoblot detection of ubiquitinated species (anti-ubiquitin), with loading controls (anti-Lyn, -Fyn, -Src or -GFP) ($n = 3$ independent enzymatic assays). Complete reagent lists, complex preparation, substrate IP conditions, wash buffers, and gel/blot parameters are provided in the Supplementary Methods and Tables.

### Generation of DLD-1 Lyn KO Flp-In T-REx cell lines and stable Lyn-GFP variants

DLD-1 Lyn knockout (KO) Flp-In T-REx cells were generated by CRISPR/Cas9 electroporation using two guide RNAs targeting the LYN locus. Editing and clone validation were confirmed by genomic sequencing, Western blotting, and MS[51]. Stable cell lines expressing doxycycline-inducible Lyn-GFP (WT, G2A, or G2P) were established by FRT/Flp-mediated recombination and validated by antibiotic selection, Western blotting, and fluorescence microscopy[52]. Detailed protocols are provided in the Supplementary Methods and Tables.

### Statistics and reproducibility

All MS experiments were performed as independent biological triplicates. Relative protein quantification was carried out using the label-free algorithm integrated in DIA-NN[53], and statistical testing was performed in Perseus using a two-sided Student's $t$ test with permutation-based multiple-testing-adjusted $p$-values[53,54], as specified in the figure legends. Data from peptide pull-downs and IPs were filtered for 3 out of 3 replicates in at least one condition, and then data imputation was performed at the lower end of the distribution (downshift: 1.8; width: 0.3). Volcano blots were visualized with RStudio (version 2023.06.0). Enzymatic activity assays were conducted with four independent assays ($n = 4$) and Michaelis–Menten kinetics calculated in GraphPad Prism (version 8.4.3). All Western blot experiments and their quantifications were performed with three independent replicates ($n = 3$). Cell experiments were performed as biological replicates, and binding assays as independent assays. NatA and different E3 ligase complexes were purified several times ($n = 3$ independent purifications). Blots were quantified in ImageJ; band intensities were normalized to the vinculin loading control, and the corresponding control condition within each blot was set to 100%. Percent values were calculated within each blot relative to its internal control. Data are presented as mean ± s.d. Fold changes are reported as the ratio of the normalized value for the condition of interest to the normalized control value, with mean ± s.d. computed across biological replicates. Uncropped blots and blots used for quantification are provided in the Source data file. Full details of software versions, parameter settings, and filtering criteria are provided in the Supplementary Methods and Tables.

### Reporting summary

Further information on research design is available in the Nature Portfolio Reporting Summary linked to this article.

## Data availability

The datasets generated during and/or analysed during the current study are available from the corresponding author. The mass spectrometry proteomics data have been deposited to the ProteomeXchange Consortium via the PRIDE partner repository with the dataset identifier PXD061095[55]. Source data are provided with this paper [https://doi.org/10.6084/m9.figshare.30885413].

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

## Acknowledgements

TB gratefully acknowledges funding by the DFG (project number: 5041 140321) and by the Friedrich-Baur Stiftung. TB and MSR are as well thankful for funding by the LMU Munich's Institutional Strategy LMUexcellent within the framework of the German Excellence Initiative and MS instrumentation by the DFG (INST 86/1800-1 FUGG). We thank Sabine Wohlgemuth and Tiziano Crocilla for their help with insect cell cultures, and Kai Walstein for support with fluorescence microscopy.

## Author contributions

T.B. designed the study and planned experiments. N.K. performed peptide pull-downs, cloning, siRNA and overexpression cell culture experiments, and high-pH fractionation. F.M. cloned and purified CUL4A-DDB1-DCAF10 complexes and NatA, and performed in vitro ubiquitination, Nt-acetylation assays, proteome fractionations, Streptavidin pull-downs and fluorescence microscopy. HN performed DLD-1 and RPE-1 experiments, generated Lyn KO cell lines overexpressing N-terminal Lyn mutants and performed their validation. L.S. performed peptide pull-downs and cell culture experiments. T.P. cloned GST-DCAF10, purified it and performed peptide binding assays. E.A. performed peptide pull-downs. J.W. performed liver peptide pulldowns. M.R. performed binding assays. I.V. performed AF3 predictions. S.M. designed the Lyn KO strategy and helped generate cell lines. T.B. and N.K. performed bioinformatic and statistical analyses. T.B., A.M., N.K., F.M., and H.N. analysed the data. T.B. wrote the manuscript draft. A.M., M.S.R., and T.B. proofread the manuscript with the help and input of all authors.

## Funding

## Competing interests

The authors declare no competing interests.
