## [Transparent Peer Review File · Nature Communications]

CUL4A-DDB1-DCAF10 is an N-recognin for N-terminally acetylated Src kinases

Corresponding Author: Professor Tanja Bange

Version 0:

Reviewer comments:

Reviewer #1

(Remarks to the Author)

The manuscript "CUL4A-DDB1-1 DCAF10 is an N-recognin for N-terminally acetylated Src kinases" by Kremer et al. establishes the molecular function of an uncharacterized Ubiquitin ligase complex. The DCAF10 recognition component mediates binding to proteins starting with an N-terminally acetylated Glycine. This makes this DCAF10 complex a rare N-recognin in the Ac/N-degron pathway which was established in 2010. This is a conceptually very important study defining a novel Ubiquitin E3 ligase recognizing and mediating the cellular degradation of N-terminally acetylated proteins, and specifically for the Src kinases which are both Nt-acetylated and myristoylated.

Comments:

1. Not sure if references are allowed in the Abstract. Please revise if this is not according to journal policies.
2. The Introduction should in general be extended to improve the coverage of relevant information and literature. For example: Lines 49-53 should include references to original literature such as Bachmair et al. Science 1986 on N-degrons, Linster et al. Nat Commun 2022 on stability of NatA substrates, Arnesen et al., PNAS 2009 and Heathcote et al., Nat Commun 2024 on specificity of NatA. The section around Line 60-63 should include information and original references on the Ac/N-degron pathway including MARCH6/TEB4/DOA10 and NOT4 E3 ligases and its function, from Hwang et al., Science 2010; Shemorry et al., Mol Cell 2013, Park et al., Science 2015.
3. Peptide pulldowns. Some of the N-terminal peptides were also significantly enriched for other ubiquitin ligases in a Nt-modification dependent manner (Supplementary data, Table S1 and S2), for example UBR5 and RNF126. Could the authors please comment of these findings, and potentially include relevant thoughts in the Discussion?
4. Figure 2 b-d. Please indicate number of repetitions and show all data in supplemental or source data. For 2b, there is a technical artefact around the Myr-Lyn band. This should be repeated to allow assessment of binding.
5. Lines 205-208: '...As Nt-acetylation and N-myristoylation compete for the same Gly, knockdown of N-myristoyltransferase 1/2 (NMT1/2) is expected to increase acetylation by NatA11,14,25. If increased Nt-acetylation leads to DCAF10 recognition, enhanced SFK degradation should follow.'
Please rewrite to already in this introduction include that reduced myristoylation could also (and will initially) lead to more N-unmodified SFK which directly could be targeted for degradation by Ub E3 ligases independent of DCAF10. (ultimately leading to the conclusions presented by the authors in Figure 3g)
6. Figure 3 and Figure 4 data should be based on 3 independent experiments (or at least key selected experiments) and also analysed statistically to assess reproducibility and significance (despite the semi-quantitative nature of Western blotting).
7. For the in vitro reconstitution of DCAF10-complex ubiquitylation of immunoprecipitated Lyn-GFP (Figure 4), why was not Lyn-GFP IPed from cells depleted for both NMT1/2 and DCAF10? This Lyn would presumably be lacking N-terminal myristoylation, having increased N-terminal acetylation, and not being degraded by cellular DCAF10-complexes.
8. Consider moving key data elements from Extended/Supplemental to main figures, for example Figure 6h. (also here,

please indicate number of independent experiments etc. in the figure legend)

9. Lines 361- "...which identified N-myristoylated DCAF10 substrates..."

Please indicate/cite the origin of the Bio-ID data.

Please rephrase. Now it reads like the DCAF10 substrate N-myristoylation is a part of the recognition, while it should state that several DCAF binders are also N-myristoylated, supporting..

Reviewer #2

(Remarks to the Author)

In this manuscript, Kremer et al. present a compelling study identifying DCAF10 as a novel substrate receptor for the CUL4A–DDB1 E3 ubiquitin ligase complex, specifically targeting N-terminally acetylated Src family kinases (SFKs). Using an elegant combination of biochemical and computational approaches—including peptide pull-down assays, mass spectrometry, AlphaFold predictions, siRNA-mediated depletion, and in vitro ubiquitination assays—the authors show that DCAF10 selectively binds N-acetylated glycine residues at the N-termini of SFKs, particularly when myristoylation is impaired. They provide strong evidence that this recognition leads to proteasomal degradation via the Ac/N-degron pathway. This mechanism is proposed to function as a co-translational quality control system, ensuring that only properly Myristoyled SFKs evade degradation. The study contributes valuable insight into the interplay between N-terminal processing, protein fate, and cellular surveillance systems. The experimental strategy is well thought out, and the use of AlphaFold to support hypothesis-driven testing is a timely and innovative feature of the work.

While this is a highly compelling and innovative manuscript that makes a strong conceptual and mechanistic contribution to the field, a few aspects would benefit from clarification or deeper exploration, as outlined below.

Major concerns:

- 1) The proposed model—where a general N-degron pathway monitors proteins starting with N-terminal glycine that fail to be myristoylated—is intriguing, but is based largely on studies involving SFKs and a few additional proteins with varied N-terminal processing used as control. In some cases, the conclusions appear to extend beyond the data. For example, while results for Lyn are consistent and convincing (e.g., decreased protein levels upon NMT1/2 knockdown), the data for Fyn are more variable: Extended Data Fig. 4a shows no decrease, Fig. 3e shows an increase in DLD-1 cells, Fig. 3f shows a decrease in RPE-1, while Figs. 3b, 3h, and 4a show no clear effect. Src is also inconsistent—unchanged in Fig. 3b but reduced in Extended Data Fig. 4a. These discrepancies should be acknowledged, quantified if possible, and interpreted with caution.
- 2) The authors refer to N-terminally acetylated SFKs as “aberrant.” However, such forms have been previously observed, albeit at lower abundance, and their function remains unclear. Unless it can be demonstrated that these forms are dysfunctional or deleterious, it may be more accurate to describe them as “non-myristoylated and acetylated” rather than “aberrant.”
- 3) In Fig. 2b, it's unclear whether a signal is present for Myr-Lyn. Improving the image quality or including a clearer explanation would help confirm that myristoylation prevents DCAF10 binding. Parallel experiments with Myr-Src and Fyn (if feasible) would further support this conclusion. It would also be helpful to explain why Fyn was not included. Signal normalization across these experiments would improve data clarity.
- 4) The discussion raises an interesting point about nuclear localization of SFKs and the potential for N-terminal acetylation to influence this process. If acetylation can lead to both degradation and nuclear targeting, it would be valuable to discuss how these outcomes are balanced or determined—e.g., is there a signal or contextual cue that directs one fate over the other?

Minor Concerns:

- 1) Lines 94–97: EF1B and TMEM97 are described as having uncharacterized N-termini, but EF1B appears annotated in Supp. Fig. S1. Please clarify.
- 2) Lines 113–114: This sentence repeats the earlier point about TMEM97 and EF1B. Consider merging or rephrasing to avoid redundancy.
- 3) Lines 176–200: NatA-dependent acetylation of Lyn and Src peptides is well documented—was Fyn tested as well? Including this would round out the analysis.
- 4) Line 207: The manuscript states that NatA-mediated acetylation increases when NMT1/2 is depleted, implying direct competition for N-terminal glycine. However, the cited references do not seem to provide direct evidence of this. Consider revising or clarifying this point.
- 5) Western Blot Migration: NMT1 and NMT2 are expected to migrate at ~50–55 kDa (per antibody datasheets), yet NMT1 appears at a higher molecular weight in the WBs. Please clarify if blots were run on the same membrane, or whether different conditions might explain the shift.
- 6) Quantification of Western Blots: For completeness, it would be helpful to include quantification for all tested proteins (including Src and Fyn), even if only in the supplementary data.
- 7) CUL2–ZYG11B Pathway: Since degradation of Lyn and Fyn by the ZYG11B/ZER1 E3 ligase is shown, it would be informative to test whether Src is also a substrate in this context.
- 8) References in Figures: Please confirm whether in-figure references are permitted under Nature Communications formatting guidelines.
- 9) Terminology Consistency: Standardize usage of terms like “N-terminal acetylation” vs. “Nt-acetylation” throughout the manuscript for clarity and consistency.
- 10) Line 151: Double-check that figure citations (e.g., Figs. 2i–j) align correctly with the main text.

11) Extended Fig. 8: The proposed model is central to the paper and would be better placed in the main figures rather than the supplementary section.

Reviewer #3

(Remarks to the Author)
NCOMMS-25-22460-T

CUL4A-DDB1-DCAF10 is an N-recognin for N-terminally acetylated Src kinases. Novel degron pathways triggered by deficient PTM N-terminal processing that can control the degradation of oncogenes (e.g. c-Src and SFKs) is an interesting subject that could have important therapeutic implications but are yet to be fully explored and exploited. In this work the authors employed peptide pull-downs, mass spectrometry, and AlphaFold 3 predictions to identify the E3 ligase adaptor DCAF10 as a receptor for aberrantly N-terminally acetylated SFKs. The authors uncover a novel N-degron pathway that supervises myristoylation replacement with acetylation and activates degradation of SFKs by DCAF10.

Major comments:

While the study is interesting and potentially important for the field, the Mass Spectrometry data for instance are of high quality and sets the working hypothesis clearly. However, the study lacks in some parts a more detailed and deep biochemical and biophysical characterization of the interacting proteins and partners identified in the study. In particular the main data have been generated by pull-downs using modified and unmodified peptides as bait and were not recapitulated with full-length or isolated domains recombinant constructs to evaluate also additional sequence constraints (allosteric control) in a full-length and more physiological context.

In addition, the characterization of the interacting interfaces between c-Src (acetylated versus myristoylated as negative control) with DCAF10 and NatA could have been dissected and the interaction and crosstalk mechanisms proposed. Also the in silico structural analyses lack biochemical or biophysical support, and structure-function relationship with functionally validated mutants were also missing, which could have been characterized using recombinant proteins and/or isolated domains by biophysical and biochemical techniques.

Minor comments (in no particular order):

1. CRISPR/Cas9-mediated knockout of endogenous LYN followed by Integration of inducible LYN-GFP variants. This is a very elegant piece of data and evidence and should be mentioned clearly in the abstract where only the siRNA data was emphasized. This would increase the impact of the paper.

2. In vitro, a CUL4A-DDB1-DCAF10 complex recapitulated ubiquitination of N34 terminally acetylated Lyn. But how does it translate to Ac-Gly starting peptides. Are the authors assuming the same mechanisms of control? Couldn't they try the same in vitro experiment with G2-acetylated protein?

3. Acetylated N-termini of Src family kinases interact directly with DCAF10
To further investigate the interaction between DCAF10 and Ac-Gly starting peptides, we focused on SFK members. All four tested Nt-acetylated peptides from this family (Lyn, Fyn, Src, and Yes) bound to DCAF10, suggesting a mechanism relevant to the entire SFK family.

Data are from pull down experiments with modified or unmodified peptides and these data were not recapitulated with intact proteins either recombinant in in vitro experiments or from a cellular context (e.g. IPs). Biophysical characterization (e.g. binding assays) for measuring affinities would be desirable. Is the same mechanism applying to all the SFKs members, or are specific and more private determinants for each of the members?

4. There are interesting in silico prediction of interactions between some modified peptides and the active site of DCAF10. However, there is no functional validation of the key interactive residues that confer specificity and drive such interactions nor any other structure-function relationship experiments. In the same line there is a need to validate such interactions between peptides (both modified and unmodified) and targets biophysically so affinities can be measurable and also provide some more structural evidence about the interacting interface and catalytic reaction.
Could the authors explore further the interaction between c-Src acetylated versus myristoylated and DCAF10 and NatA and map the interaction interfaces and mechanism of regulation?

5. In the in vitro ubiquitination assays. Did the author look at the targeted lysine residues and ID them by MS? Could this bring any insights into the mechanism of interaction and regulation?

6. What is the functional role of acetylated versus myristoylated c-Src and SFKs? Are they functionally distinct in terms of phospho-tyrosine activity?

7. What is known about N-degron pathways in cancer and other human disorders? What is the therapeutic potential of this PTM quality control system? How could it be used therapeutically? Perhaps the author should make emphasis in the discussion on these questions.

Version 1:

Reviewer comments:

Reviewer #1

(Remarks to the Author)

All concerns have been addressed appropriately.

Reviewer #2

(Remarks to the Author)

The authors have provided clear and satisfactory responses to all my previous comments. They have added the missing Fyn data, standardized quantifications (n = 3), expanded the mechanistic analyses with full-length IP–MS experiments and a DCAF10 pocket mutant, and addressed all minor editorial points. I am satisfied that they have fully addressed my comments.

The revisions strengthen the manuscript without overextending its scope. I have no remaining concerns and recommend acceptance in its current form.

Reviewer #3

(Remarks to the Author)

Revised manuscript NCOMMS-25-22460A

The authors have presented an improved version of the manuscript that answer most of the questions raised by the reviewers. It is clear the authors have tried to provide a more detailed and deeper biochemical analyses of the interacting proteins and partners and their mechanism of action in the study at a cellular level. I acknowledge also that getting N-terminal modified recombinant proteins (SFKs members) is a difficult and challenging task and may not be feasible in the time frame of the revision. I assume this is the reason also why they have chosen the pull-down strategy with modified peptides for MS. Regarding the data presented with the full-length proteins in a cellular context I still have some comments that need to be answered and clarified. Mainly, about the interpretation of data from Figure 4 and 6. Besides I have a couple of technical questions that called my attention as well.

Figure 1. In panel g, author show a clear positive correlation between DCAF10 and acetylated-Gly and not with myristoylated-Gly peptides of Lyn. However, such experimental data is not recapitulated with the other SFKs family members peptides. The authors show however volcano plots with the interacting profile of DCAF10 in presence of acetylated-Gly or unmodified N-terminal peptides from SFKs members. Could the author comment on this please?

Figure 2. In panel m-p authors show WBs with GST-DCAF10[aa 120-559] which runs in gel at 100 kDa. This size seems to me rather high, compared with data shown on Figure 5b related to MBP-DCAF10 (full-length??) which runs at the same Mw (103 kDa), having in mind the significant difference in size between both tags used (GST-26kDa and MBP-42kDa). Could the authors comment on this please?

Besides, it is not clear what are the interpretation of the WBs from panel m-p in terms of expected bands and interactions. Could the authors comment on this please?

In addition, the authors show docking data however no MD simulation (300-400 ns) is provided to test the stability and thermodynamic parameters of the interaction. I think the peptide should be shown on sticks and the interactions depicted explicitly. These data is important and will add support to the 3D-docking prediction from AF3.

Figure 3. Did the author used N-Myr-peptides as negative controls? Did they also try to monitor enzymatic activity of NatA in presence of DCAF10 towards N-Acetylated SFKs peptides?

Figure 4. Panel a and b. Proposed model 1. N-Myr-Gly protein is viable no degron-mediated degradation. 2. In NMT1/2 deficient cells (depleted) acetylated N-term-Gly SFKs will be degraded via DCAF10 degron. 3. In DCAF10 depleted cells N-Myr-Gly SFKs members, as expected, do not undergo degradation. 4. However, In DCAF10 depleted cells, if NMT1/2 is also depleted N-Acet-Gly proteins will not underdo degradation.

Under these premises, we should expect a drop in the protein levels of all SFKs members under condition 2. However, this is only evident on Lyn and not in c-Src nor Fyn cases. Could the author comment on this please?

In addition, in DCAF10 RNAi cells, I see in some cases that upon depletion of NMT1/2 (hence, favoring N-Acetylated Gly in SFKs) there is a decrease (e.g. Lyn) in some cases and a significant increase (e.g. c-Src) in others, and I do not fully understand these differences as the patten should be consistent across the distinct SFKs family members. Could the author comment on this please?

On panel j, why author show MS data and not WBs or qPCR of DCAF10?

Figure 5. The new figure 5 is labelled as Figure 4.

What does the Ub signal above the 250 kDa band correspond to exactly? Should we expect a prominent band/signal around the size of SFKs (50-60 kDa) which at the same time it will correlate with lower levels of total SFKs members, and viceversa, when Ub is diminished, increased SFKs levels should be seen. If the authors could clarify this

Figure 6: On panel a, would the anti-Lyn antibody recognize the ectopic GFP-tagged form of Lyn as well, just as internal control? Why did the authors not show this data?

It would have been nice to see the DLD-1 Lyn KO Flp-In T-REx Lyn-GFP system applied to other SFKs family members.

Figure 7: The new Figure 7 is labelled as Extended Data Figure 8 Kremer et al.

REVIEWER COMMENTS

We thank all three reviewers for their constructive feedback. The revisions and new data substantially strengthen our conclusions and clarify both the shared features and the distinct behaviors of Lyn, Fyn, and Src with respect to DCAF10 recognition. Below, we provide our point-by-point responses to each comment.

Reviewer #1 (Remarks to the Author):

The manuscript “CUL4A-DDB1-1 DCAF10 is an N-recognin for N-terminally acetylated Src kinases” by Kremer et al. establishes the molecular function of an uncharacterized Ubiquitin ligase complex. The DCAF10 recognition component mediates binding to proteins starting with an N-terminally acetylated Glycine. This makes this DCAF10 complex a rare N-recognin in the Ac/N-degron pathway which was established in 2010. This is a conceptually very important study defining a novel Ubiquitin E3 ligase recognizing and mediating the cellular degradation of N-terminally acetylated proteins, and specifically for the Src kinases which are both Nt-acetylated and myristoylated.

Comments:

1. Not sure if references are allowed in the Abstract. Please revise if this is not according to journal policies.

We thank the reviewer. We have removed all references from the Abstract to comply with the journal's policy.

2. The Introduction should in general be extended to improve the coverage of relevant information and literature. For example: Lines 49-53 should include references to original literature such as Bachmair et al. Science 1986 on N-degrons, Linster et al. Nat Commun 2022 on stability of NatA substrates, Arnesen et al., PNAS 2009 and Heathcote et al., Nat Commun 2024 on specificity of NatA. The section around Line 60-63 should include information and original references on the Ac/N-degron pathway including MARCH6/TEB4/DOA10 and NOT4 E3 ligases and its function, from Hwang et al., Science 2010; Shemorry et al., Mol Cell 2013, Park et al., Science 2015.

We appreciate the reviewer's constructive suggestion. The Introduction has been revised to provide a more complete overview of N-degron and Ac/N-degron pathways and to cite the original literature as recommended. Specifically:

- We now reference the foundational discovery of the N-degron pathway (*Bachmair et al., Science 1986*) and its physiological importance in NatA substrate stability (*Linster et al., Nat Commun 2022; Arnesen et al., PNAS 2009; Heathcote et al., Nat Commun 2024*).
- The section describing acetylation-dependent degradation has been expanded to include key studies establishing the Ac/N-degron pathway and its E3 ligases (*Hwang et al., Science 2010; Shemorry et al., Mol Cell 2013; Park et al., Science 2015*), together with MARCH6/TEB4 and NOT4.

These additions now appear in the revised Introduction (paragraphs 2 and 3).

3. Peptide pull-downs. Some of the N-terminal peptides were also significantly enriched for other ubiquitin ligases in a Nt-modification dependent manner (Supplementary data, Table S1 and S2), for example UBR5 and RNF126. Could the authors please comment of these findings, and potentially include relevant thoughts in the Discussion?

We appreciate the reviewer's careful observation. In our peptide pull-downs, we indeed detected additional RING- and HECT-type E3s (e.g., UBR4/5, UBR2, RNF126). To nominate *bona fide* N-recognin candidates, we applied stringent filters: (i) significant enrichment between matched peptide pairs ($p < 0.05$; fold-change > 2), (ii) reproducibility across biological replicates and across peptides sharing key sequence features, and (iii) relevance to a coherent substrate class (SFKs) where the modification is physiologically plausible. DCAF10 consistently met these criteria and was therefore pursued with orthogonal validation (direct binding with GST-DCAF10, in vitro ubiquitination by CUL4A-DDB1-DCAF10, and cellular genetics). In contrast, UBR4 and RNF126 were frequently detected but did not pass significance or fold change thresholds; UBR5 often showed $p < 0.05$ for binding to the Nt-free peptide but lacked > 2 -fold enrichment and consistent modification-dependent behavior; UBR2 appeared only sporadically for Gly-starting peptides. Among these, UBR5 appears to be an interesting candidate that may merit targeted follow-up. However, without applying our full validation pipeline, we are cautious with the interpretation of MS-only signals, and non-significant binding may reflect nonspecific bead interactions.

Given the scope and length of the manuscript, and because we did not perform further mechanistic validation for these additional E3s, we have not expanded the Discussion. We prefer to keep the narrative focused on experimentally validated DCAF10-dependent Ac-Gly recognition. Looking ahead, it will be important to test how many N-recognin candidates can be validated, how many N-degron surveillance pathways exist, and to what extent their substrate specificities overlap or partition under different cellular conditions.

4. Figure 2 b-d. Please indicate number of repetitions and show all data in supplemental or source data. For 2b, there is a technical artefact around the Myr-Lyn band. This should be repeated to allow assessment of binding.

We agree that the myristoylated (Myr) peptide pull-down blots showed artefactual behavior. Despite repeating the experiment and extensively optimizing buffer composition, salt, detergents, blocking reagents, and incubation/wash conditions, Myr-peptides behaved variably in our bead-based assay; consistent with hydrophobic long-chain effects (aggregation/micelle formation and nonspecific adsorption). Because we concluded that this assay format is not suitable to demonstrate lack of binding for Myr peptides, we removed the Myr panels from Fig. 2.

To obtain an orthogonal, quantitative readout, we made a concerted effort to purify DCAF10 at high yield and purity (different constructs, buffers, etc.) to attempt equilibrium binding measurements (e.g., SPR to estimate apparent K_d values for Ac- vs Myr-peptides). However, in our hands recombinant DCAF10,

when isolated without its CUL4A–DDB1 complex partners, proved technically challenging to handle (tendency to instability/aggregation), and we did not achieve a reliable, quantitative binding assay.

In the revised figures, we now (i) include Fyn (Ac- and Nt-free peptides) alongside Lyn, Src and THOC7 (**new Fig. 2m–p**); (ii) standardize all peptide pull-downs to $n = 3$ biological replicates, and (iii) report binding as mean \pm s.d. normalized to the Ac-peptide (**new Supplementary Fig. 4b**). We also ran Ac-Lyn, Ac-Fyn, and Ac-Src side-by-side ($n = 3$) to compare relative binding (**Supplementary Fig. 4c–d**), showing Lyn ($\approx 100\%$) \approx Fyn ($94 \pm 6\%$) $>$ Src ($70 \pm 16\%$). All underlying blots are provided in the Source Data.

We acknowledge that we cannot ultimately show lack of binding of the Myr peptide, but note that we have good supporting evidence that Myr peptides are unlikely to engage: (i) AlphaFold 3 modeling predicts absence of pocket engagement and we further validated the prediction with a triple DCAF10 mutant which abolished ubiquitination of Lyn and Fyn but not Src (**new Figure 5i–k**), and (ii) in our peptide pull-downs, Myr-Src shows no detectable DCAF10 binding, while Myr-Lyn/Myr-Fyn signals are $\sim \log_2$ 5–6 lower than their acetylated counterparts (raw values in the new **Supplementary Fig. 1k**). (iv) Lyn^{WT}-GFP binds to DCAF10 (**new Fig. 6j–l**) but not N-terminal mutants.

5. Lines 205-208: ‘..As Nt-acetylation and N-myristoylation compete for the same Gly, knockdown of N-myristoyltransferase $\frac{1}{2}$ (NMT1/2) is expected to increase acetylation by NatA11,14,25. If increased Nt-acetylation leads to DCAF10 recognition, enhanced SFK degradation should follow.’ Please rewrite to already in this introduction include that reduced myristoylation could also (and will initially) lead to more N-unmodified SFK which directly could be targeted for degradation by Ub E3 ligases independent of DCAF10. (ultimately leading to the conclusions presented by the authors in Figure 3g)

We thank the reviewer for this comment and agree that the two possible glycine fates/modifications and routes should be addressed here. We revised the paragraph as follows: “Consistent with co-translational action of NMT and NatA on overlapping Nt-substrates, lowering NMT1/2 is expected to bias nascent SFKs toward mutually exclusive fates—Nt-acetylated or Nt-free—routing them either to CUL4A-DDB1-DCAF10 or to the published CUL2-Elongin B/C-ZYG11B/ZER1 Gly/N-degron pathway, respectively”.

For clarity, we respectfully note that our model does not imply a sequential “initially unmodified \rightarrow then acetylated” process. Rather, co-translational processing at the ribosome yields parallel, mutually exclusive Nt-states (Nt-free vs. Nt-acetylated) that subsequently engage distinct degradation branches (as schematically shown in **Fig. 4i**).

6. Figure 3 and Figure 4 data should be based on 3 independent experiments (or at least key selected experiments) and also analysed statistically to assess reproducibility and significance (despite the semi-quantitative nature of Western blotting).

We agree with the reviewer on the importance of reproducibility. In the revised manuscript, key Western-blot experiments for Lyn, Fyn, Src, and THOC7 were repeated as three independent biological replicates

(n = 3), and we report individual data points, mean \pm s.d., and fold changes (new **Fig. 4b–f, 4j–m**; Source Data). Conditions were kept as constant as possible within each experiment (same antibody lots/dilutions, identical imaging parameters, linear non-saturated quantification, and normalization to the same loading control). Given the semi-quantitative nature of Western blotting and additional variance introduced by 72 h siRNA in asynchronously cycling cells, we present the data descriptively.

Per the reviewer's request, we also performed statistical testing on the underlying n = 3 datasets. As expected for semi-quantitative WB measurements with small effect sizes or higher variability, most comparisons are not significant, with the exception of selected Lyn conditions and fold-change contrasts. We emphasize a quantitative description (mean \pm s.d. and fold change) in the main text and hope the reviewer agrees this is the most accurate and transparent representation of these data.

7. For the *in vitro* reconstitution of DCAF10-complex ubiquitylation of immunoprecipitated Lyn-GFP (Figure 4), why was not Lyn-GFP IPed from cells depleted for both NMT1/2 and DCAF10? This Lyn would presumably be lacking N-terminal myristoylation, having increased N-terminal acetylation, and not being degraded by cellular DCAF10-complexes.

We appreciate this suggestion. Our earlier *in vitro* assays with endogenous Lyn did not reveal clear differences between RNAi treated and control cells (**Fig. 5c, e, g**) and for practical reasons we performed the experiments with untreated lysates. We followed the reviewer's advice and performed new *in vitro* ubiquitination assays comparing Lyn^{WT}-GFP IPs from NMT1/2 siRNA-treated versus control cells using the three purified ligase complexes (WT E3 complex, catalytic RBX1^{mut}, and -DCAF10). We observed a robust increase (>5-fold) in ubiquitination with the WT complex for the NMT1/2-depleted substrate, with no increase for RBX1^{mut} or -DCAF10 controls—fully consistent with our model (new **Fig. 6g**). We interpret this as a substrate-availability effect: under NMT1/2 knockdown, a larger fraction of Lyn-GFP is Nt-acetylated while endogenous DCAF10 is limiting (whole-proteome MS confirms low expression), allowing an acetylated pool to accumulate in cells and be efficiently modified *in vitro* when the reconstituted DCAF10 complex is supplied. We repeated these assays with all N-terminal variants (WT, G2A, G2P) in three independent biological replicates; results are shown in new **Fig. 6h–i**. For the assays using control lysates, we performed an additional replicate to reach n=3 and moved these data to **Supplementary Fig. 6h–i**. We thank the reviewer for prompting this experiment, which further strengthens our conclusions.

8. Consider moving key data elements from Extended/Supplemental to main figures, for example Figure 6h. (also here, please indicate number of independent experiments etc. in the figure legend)

We thank the reviewer for this suggestion. We have moved the former Supplementary Fig. 6h to the main figures (new **Fig. 6e**), quantified three independent biological replicates (new **Fig. 6f**), indicated n in the figure legend, and provided all underlying Western blots in the Source Data.

9. Lines 361- “..which identified N-myristoylated DCAF10 substrates...”
Please indicate/cite the origin of the Bio-ID data.
Please rephrase. Now it reads like the DCAF10 substrate N-myristoylation is a part of the recognition, while it should state that several DCAF binders are also N-myristoylated, supporting..

We thank the reviewer for this comment. We have corrected the citation and attribute the BioID dataset to Raisch *et al.* (MCP, 2023). We rephrased the sentence to: “Proximity-labeling (BioID) data identify several DCAF10-associated proteins that are typically N-myristoylated, including PPM1G, PPM1B, ARL3, DDX46, FAM49B, and Yes/Src, suggesting a potential intersection between DCAF10 interactors and N-myristoylated proteins. Importantly, our binding and modeling data suggest that DCAF10 recognizes N-terminally Ac-Gly rather than Myr-Gly per se.”

Reviewer #2 (Remarks to the Author):

In this manuscript, Kremer et al. present a compelling study identifying DCAF10 as a novel substrate receptor for the CUL4A–DDB1 E3 ubiquitin ligase complex, specifically targeting N-terminally acetylated Src family kinases (SFKs). Using an elegant combination of biochemical and computational approaches—including peptide pull-down assays, mass spectrometry, AlphaFold predictions, siRNA-mediated depletion, and *in vitro* ubiquitination assays—the authors show that DCAF10 selectively binds N-acetylated glycine residues at the N-termini of SFKs, particularly when myristoylation is impaired. They provide strong evidence that this recognition leads to proteasomal degradation via the Ac/N-degron pathway. This mechanism is proposed to function as a co-translational quality control system, ensuring that only properly Myristoyled SFKs evade degradation. The study contributes valuable insight into the interplay between N-terminal processing, protein fate, and cellular surveillance systems. The experimental strategy is well thought out, and the use of AlphaFold to support hypothesis-driven testing is a timely and innovative feature of the work.

While this is a highly compelling and innovative manuscript that makes a strong conceptual and mechanistic contribution to the field, a few aspects would benefit from clarification or deeper exploration, as outlined below.

Major concerns:

1) The proposed model—where a general N-degron pathway monitors proteins starting with N-terminal glycine that fail to be myristoylated—is intriguing, but is based largely on studies involving SFKs and a few additional proteins with varied N-terminal processing used as control. In some cases, the conclusions appear to extend beyond the data. For example, while results for Lyn are consistent and convincing (e.g., decreased protein levels upon NMT1/2 knockdown), the data for Fyn are more variable: Extended Data Fig. 4a shows no decrease, Fig. 3e shows an increase in DLD-1 cells, Fig. 3f shows a decrease in RPE-1, while Figs. 3b, 3h, and 4a show no clear effect. Src is also inconsistent—unchanged in Fig. 3b but reduced in Extended Data Fig. 4a. These discrepancies should be acknowledged, quantified if possible, and interpreted with caution.

We thank the reviewer and agree on a cautious interpretation and have revised the manuscript to avoid overstatements. Our mechanistic evidence primarily centers on SFKs, and we explicitly acknowledge this scope in the revised *Discussion*: “Although our mechanistic evidence focuses on SFKs, DCAF10 preferentially binds Nt-acetylated N-termini from several typically myristoylated proteins, suggesting broader relevance that remains to be established. Pending proteome-wide *in vivo* validation, we

provisionally refer to this DCAF10 branch as the Ac-Gly/MO (myristoylation-omitted) N-degron pathway (Fig. 7a-b).”

We also agree that the magnitude of effects varies across SFKs. Lyn shows the most prominent and consistent response, whereas Fyn and Src display smaller or context-dependent changes in steady-state amounts. We thank the reviewer for prompting additional repetitions; these clarified both shared and distinct behaviors among SFKs. In the revision, all key experiments were repeated as three independent biological replicates ($n = 3$) and quantified for Lyn, Fyn, Src, and THOC7, with individual data points, mean \pm s.d., and fold changes reported (figures and legends indicate n ; raw blots and quantifications are provided in the Source Data). Throughout, we maintain a quantitative, descriptive presentation, acknowledge differences, and avoid over-interpretation.

To contextualize variability, we note that Western-blot densitometry is semi-quantitative and technically sensitive (e.g., acquisition/exposure, membrane handling), and that 72 h siRNA in asynchronously cycling cells can introduce biological adaptations that differ between experiments and cell types. We minimized these factors by holding conditions constant within experiments (same antibody lots/dilutions, parallel gels/transfers, identical imaging parameters, linear non-saturated quantification, and a shared loading control), and we therefore emphasize effect size (fold change) and mean \pm s.d. rather than formal hypothesis testing.

2) The authors refer to N-terminally acetylated SFKs as “aberrant.” However, such forms have been previously observed, albeit at lower abundance, and their function remains unclear. Unless it can be demonstrated that these forms are dysfunctional or deleterious, it may be more accurate to describe them as “non-myristoylated and acetylated” rather than “aberrant.”

We absolutely agree and thank the reviewer for this comment. The term “aberrant/incorrect(ly)” could imply dysfunction, which we do not demonstrate. We have removed this terminology throughout and now use neutral descriptors such as “non-myristoylated, Nt-acetylated (Ac-Gly) SFKs” and “alternative N-termini.” In the course of the revision, additional experiments strengthened this nuance e.g. DCAF10 RNAi has small effects in control conditions. We have updated the *Discussion* to acknowledge these observations, to emphasize that their function remains to be established, and to avoid over-interpretation.

3) In Fig. 2b, it's unclear whether a signal is present for Myr-Lyn. Improving the image quality or including a clearer explanation would help confirm that myristoylation prevents DCAF10 binding. Parallel experiments with Myr-Src and Fyn (if feasible) would further support this conclusion. It would also be helpful to explain why Fyn was not included. Signal normalization across these experiments would improve data clarity.

We agree that the myristoylated (Myr) peptide pull-down blots showed artefactual behavior. Despite repeating the experiment and extensively optimizing buffer composition, salt, detergents, blocking reagents, and incubation/wash conditions, Myr-peptides behaved variably in our bead-based assay; consistent with hydrophobic long-chain effects (aggregation/micelle formation and nonspecific adsorption). Because we concluded that this assay format is not suitable to demonstrate lack of binding for Myr peptides, we removed the Myr panels from Fig. 2.

To obtain an orthogonal, quantitative readout, we made a concerted effort to purify DCAF10 at high yield and purity (different constructs, buffers, etc.) to attempt equilibrium binding measurements (e.g., SPR to estimate apparent K_d values for Ac- vs Myr-peptides). However, in our hands recombinant DCAF10, when isolated without its CUL4A–DDB1 complex partners, proved technically challenging to handle (tendency to instability/aggregation), and we did not succeed in obtaining a reliable quantitative binding assay.

Fyn was not included in the initial submission because we expected it to behave similarly to Lyn, but we recognize that we need experimental data and added Fyn. In the revised figures, we now (i) include Fyn (Ac- and Nt-free peptides) alongside Lyn, Src, and THOC7 (new Fig. 2m–p); (ii) standardize all peptide pull-downs to $n = 3$ biological replicates; and (iii) report mean \pm s.d. normalized to the Ac-peptide (new Supplementary Fig. 4b). We also ran Ac-Lyn, Ac-Fyn, and Ac-Src side-by-side ($n = 3$) to compare relative binding (Supplementary Fig. 4c–d), yielding Lyn ($\approx 100\%$) \approx Fyn ($94 \pm 6\%$) $>$ Src ($70 \pm 16\%$). All underlying blots are provided in the Source Data.

We acknowledge that we cannot directly demonstrate lack of binding for the Myr peptide in our assay. Nonetheless, several observations support that Myr peptides are unlikely to engage DCAF10: (i) AlphaFold 3 modeling predicts absence of pocket engagement, and a triple DCAF10 pocket mutant abolishes ubiquitination of Lyn and Fyn but not Src (new Fig. 5i–k); (ii) in peptide pull-downs, Myr-Src shows no detectable DCAF10 binding, while Myr-Lyn/Myr-Fyn signals are $\sim \log_2$ 5–6 lower than their acetylated counterparts (raw values in new Supplementary Fig. 1k); and (iii) full-length Lyn^{WT}–GFP binds DCAF10 (new Fig. 6j–l), whereas N-terminal mutants do not.

Together, these data support the conclusion that Ac-Gly is required for robust DCAF10 engagement, while we refrain from asserting that myristoylation per se prevents binding.

4) The discussion raises an interesting point about nuclear localization of SFKs and the potential for N-terminal acetylation to influence this process. If acetylation can lead to both degradation and nuclear targeting, it would be valuable to discuss how these outcomes are balanced or determined—e.g., is there a signal or contextual cue that directs one fate over the other?

We agree this is an intriguing point. In the revised Discussion, we first clarify scope—our data support an Ac-Gly/MO branch for SFKs—and then acknowledge a potential role for N-terminal acetylation in shaping SFK localisation alongside degradation. In principle, the balance could reflect (i) co-translational partitioning of nascent chains into Nt-acetylated versus Nt-free species (NatA vs NMT), (ii) availability and specificity of Ac/N-recognins (e.g., DCAF10 vs ZYG11B/ZER1) together with proximal sequence context near Gly2, and (iii) exposure of basic motifs/NLS-like elements and nuclear export. While co-translational processing is increasingly recognised as dynamic and regulatable (e.g., nascent-chain length, ribosome-associated factors, stress conditions), to our knowledge there are no published data demonstrating cue-dependent modulation of the specific choice between co-translational myristoylation and N-terminal acetylation, nor its routing into distinct N-degron branches. Accordingly, to avoid becoming too speculative, we limit our discussion to observations supported by our data and will address these decision cues in future work.

Minor Concerns:

1) Lines 94–97: EF1B and TMEM97 are described as having uncharacterized N-termini, but EF1B appears annotated in Supp. Fig. S1. Please clarify.

We thank the reviewer for this comment. EF1B is not annotated as Nt-myristoylated in UniProt and we changed Supplementary Fig. S1 accordingly.

2) Lines 113–114: This sentence repeats the earlier point about TMEM97 and EF1B. Consider merging or rephrasing to avoid redundancy.

We rephrased slightly to avoid redundancy: “Among the nine DCAF10-binding peptides, seven correspond to proteins known to be N-myristoylated, while two (TMEM97 and EF1B) have not been characterized as such.”

3) Lines 176–200: NatA-dependent acetylation of Lyn and Src peptides is well documented—was Fyn tested as well? Including this would round out the analysis.

We agree with the reviewer and thank them for the suggestion. We have repeated the *in vitro* acetylation experiments using both Lyn and Fyn peptides and determined Michaelis–Menten kinetics for Fyn, revealing a K_m value of 235 μM , which lies between those of Src (143 μM) and Lyn (411 μM). This result confirms that Fyn is also a NatA substrate, with an acetylation efficiency comparable to the other SFK members.

4) Line 207: The manuscript states that NatA-mediated acetylation increases when NMT1/2 is depleted, implying direct competition for N-terminal glycine. However, the cited references do not seem to provide direct evidence of this. Consider revising or clarifying this point.

We thank the reviewer for this important clarification. We have rephrased the text throughout to avoid implying proof of direct competition. We now state that the observed increase in NatA-mediated acetylation upon NMT1/2 depletion is consistent with (but does not by itself prove) co-translational competition at Gly-starting N-termini. This interpretation is supported by ribosome-level structural and mechanistic work (e.g., Deuerling/Ban/Shan and colleagues) showing that NAC can co-recruit NMT, METAP1, and NatA to the tunnel exit, positioning these enzymes in close proximity and making dual access to nascent Gly plausible.

5) Western Blot Migration: NMT1 and NMT2 are expected to migrate at ~50–55 kDa (per antibody datasheets), yet NMT1 appears at a higher molecular weight in the WBs. Please clarify if blots were run on the same membrane, or whether different conditions might explain the shift.

In our workflow, we typically probed NMT1 first, stripped, and then reprobed for NMT2; in some experiments they were run on separate membranes. Across all experiments, NMT1 reproducibly migrates slightly above NMT2 and above its predicted mass. We observe minor differences attributable to gel format (e.g., 10% in-house gels vs. commercial gradient gels), but no condition-dependent size shifts; the relative spacing (NMT1 > NMT2) is always consistent. Uncropped blots in the Source Data (e.g., **Fig. 6d**, ladder

with 50 and 75 kDa markers) show NMT1 and NMT2 on different blots run together, where NMT1 runs slightly higher. The band identity is unambiguous, although we do not currently have an explanation for the higher apparent molecular weight of NMT1. This behavior is also compatible with vendor controls, which report NMT1 migrating between 50–70 kDa. (<https://www.abcam.com/en-us/products/primary-antibodies/nmt1-nmt-antibody-ab186123>).

Fig. 1 for reviewers only: Comparison of uncropped blots anti-NMT1 and anti-NMT2

6) Quantification of Western Blots: For completeness, it would be helpful to include quantification for all tested proteins (including Src and Fyn), even if only in the supplementary data.

We agree with the reviewer. We now provide quantifications for Lyn, Fyn, and Src ($n = 3$) with mean \pm s.d. and individual data points in the main figures (**Fig. 4b–f**). Presenting these data in the main figures highlights both the shared trends and distinct differences among the three SFKs, which we consider important for interpreting their relative behavior. The corresponding raw blots are included in the Source Data.

7) CUL2–ZYG11B Pathway: Since degradation of Lyn and Fyn by the ZYG11B/ZER1 E3 ligase is shown, it would be informative to test whether Src is also a substrate in this context.

We agree with the reviewer. We added a third replicate for Lyn and Fyn and performed three independent replicates for Src, with corresponding quantifications (**Fig. 4j–m**; mean \pm s.d.). These data reveal member-specific routing within the SFK family: Lyn and Fyn show greater dependence on DCAF10, whereas Src is

more sensitive to ZYG11B/ZER1 depletion. Notably, combined perturbation of both branches yields additive stabilization across all three SFK members, consistent with complementary surveillance by the DCAF10 (Ac-Gly/MO) and ZYG11B/ZER1 (Gly/N-degron) pathways. We discuss these differences in the Results and Discussion sections.

8) References in Figures: Please confirm whether in-figure references are permitted under Nature Communications formatting guidelines.

We thank the reviewer for this remark. We have removed the in-figure citations.

9) Terminology Consistency: Standardize usage of terms like “N-terminal acetylation” vs. “Nt-acetylation” throughout the manuscript for clarity and consistency.

We thank the reviewer for the comment and have revised to ensure consistent use of “Nt-acetylation”.

10) Line 151: Double-check that figure citations (e.g., Figs. 2i–j) align correctly with the main text.

We thank the reviewer for this comment and revised to correct figure citations.

11) Extended Fig. 8: The proposed model is central to the paper and would be better placed in the main figures rather than the supplementary section.

We thank the reviewer for this helpful suggestion. We have moved the proposed model from Extended Data Fig. 8 to the main figure panel, where it now appears as main **Fig. 7**.

Reviewer #3 (Remarks to the Author):

NCOMMS-25-22460-T

CUL4A-DDB1-DCAF10 is an N-recognin for N-terminally acetylated Src kinases.

Novel degron pathways triggered by deficient PTM N-terminal processing that can control the degradation of oncogenes (e.g. c-Src and SFKs) is an interesting subject that could have important therapeutic implications but are yet to be fully explored and exploited. In this work the authors employed peptide pull-downs, mass spectrometry, and AlphaFold 3 predictions to identify the E3 ligase adaptor DCAF10 as a receptor for aberrantly N-terminally acetylated SFKs. The authors uncover a novel N-degron pathway that supervises myristoylation replacement with acetylation and activates degradation of SFKs by DCAF10.

Major comments:

While the study is interesting and potentially important for the field, the Mass Spectrometry data for instance are of high quality and sets the working hypothesis clearly. However, the study lacks in some

parts a more detailed and deep biochemical and biophysical characterization of the interacting proteins and partners identified in the study. In particular the main data have been generated by pull-downs using modified and unmodified peptides as bait and were not recapitulated with full-length or isolated domains recombinant constructs to evaluate also additional sequence constraints (allosteric control) in a full-length and more physiological context.

In addition, the characterization of the interacting interfaces between c-Src (acetylated versus myristoylated as negative control) with DCAF10 and NatA could have been dissected and the interaction and crosstalk mechanisms proposed. Also the *in silico* structural analyses lack biochemical or biophysical support, and structure-function relationship with functionally validated mutants were also missing, which could have been characterized using recombinant proteins and/or isolated domains by biophysical and biochemical techniques.

We agree that moving beyond peptide pull-downs to more physiological and mechanistic assays is important. In the revision, we therefore added experiments to increase mechanistic depth, map sequence constraints, and establish interaction in a full-length cellular setting:

1. Full-length interaction in cells (IP-MS/Co-IP): Immunoprecipitation of full-length Lyn^{WT}-GFP from cells \pm NMT1/2 siRNA, analyzed by mass spectrometry, demonstrates interaction with DCAF10 in cells. N-terminal mutants did not bind, and NMT1/2 siRNA increased DCAF10 association relative to control (**Fig. 6j-l**).
2. Local sequence constraints near the N-terminus (~positions 5–7)
Swap constructs (THOC7 acidic segment \leftrightarrow Lyn basic segment) indicate that local context biases DCAF10 recognition and that acidic residues impede binding (Fig. 2h-j; Suppl. Fig. 3e-f). “AAA” neutralization mutants (Lyn) maintain binding, consistent with a tolerance for neutral side chains within this window. Overall, acidic residues at positions 5–7 reduce engagement, whereas myristoylation-compatible features (basic/hydrophobic character and positive charge near positions 6–7) favour binding.
3. DCAF10 pocket—structure—function specificity:
Guided by AlphaFold 3, a DCAF10 triple pocket mutant abolishes ubiquitination of Lyn and Fyn but not Src, supporting the predicted Ac-Gly binding pocket and directly linking pocket integrity to engagement and ubiquitination (**Fig. 5i-k**).
4. AF3 predictions suggest that substituting Src Asn4 \rightarrow Ile permits deeper accommodation of the N-terminus in the pocket, placing Src comparably to Fyn and Lyn in the modelled pose, consistent with a preference for a hydrophobic residue at position 4 (**Supplementary Fig. 9**).
We note this as a structural hypothesis; experimental validation remains to be done.

Reproducibility and quantification: All key experiments were repeated as $n = 3$ biological replicates with quantitative reporting (individual data points, mean \pm s.d., fold change; e.g., main **Figs. 2m-p, 4b-f, 4j-m; Suppl. Figs. 4b-d, 5**), enabling direct comparison of shared features and member-specific differences in DCAF10 recognition across SFKs, which we discuss in the Results and Discussion.

We also note the reviewer's point regarding the potential crosstalk between NatA and DCAF10. We fully agree that this is an important and intriguing direction to pursue. However, addressing this mechanistically will require extensive structural and biochemical work—including the generation of stable NatA–ribosome–substrate complexes and recombinant reconstitution—beyond the scope of the present study. At this stage, available data in the literature are insufficient to propose a detailed mechanistic model, and we therefore leave it as an open question for future investigation.

Minor comments (in no particular order):

1. CRISPR/Cas9-mediated knockout of endogenous LYN followed by Integration of inducible LYN-GFP variants. This is a very elegant piece of data and evidence and should be mentioned clearly in the abstract where only the siRNA data was emphasized. This would increase the impact of the paper.

We thank the reviewer for this suggestion. We have revised the Abstract to explicitly mention the CRISPR/Cas9-mediated knockout of endogenous Lyn and the integration of inducible Lyn–GFP variants, in addition to the siRNA experiments. We agree this strengthens the Abstract.

2. *In vitro*, a CUL4A-DDB1-DCAF10 complex recapitulated ubiquitination of N terminally acetylated Lyn. But how does it translate to Ac-Gly starting peptides. Are the authors assuming the same mechanisms of control? Couldn't they try the same *in vitro* experiment with G2-acetylated protein?

We thank the reviewer and agree that direct biochemical validation using an Ac-Gly–modified SFK would represent the cleanest way to demonstrate binding specificity. However, obtaining sufficient amounts of homogeneous Ac-Gly–modified SFKs is technically challenging. Insect-cell expression might yield a small fraction of Nt-acetylated glycine species (and we currently cannot quantify this fraction), and bacterial expression produces Nt-free Gly (bacteria do not perform Nt-acetylation) that cannot be efficiently acetylated *in vitro*. Moreover, post-purification acetylation does not allow us to separate acetylated from unmodified protein to obtain a pure Ac-Gly pool for assays. Production of site-specifically acetylated proteins by synthetic peptide ligation or cell-free incorporation would be possible in principle, but currently prohibitively expensive for full-length SFKs. Cell-based strategies to enrich Ac-Gly variants via NatA overexpression are also limited, as NatA levels are tightly controlled; both strong knockdown and overexpression lead to cell death due to the essential, dosage-sensitive nature of NatA. Given these experimental constraints, we have not yet identified a feasible setup to produce a fully acetylated Gly-starting SFK. We therefore used N-terminal mutants (e.g., G2A and G2P) as the best available proxies to dissect the role of N-terminal state and modification. These data support that both an N-terminal glycine and its acetylation are required for DCAF10-dependent recognition. We fully agree that developing a strategy to generate pure Ac-Gly–modified SFKs will be an important goal for future biochemical and structural studies.

3. Acetylated N-termini of Src family kinases interact directly with DCAF10. To further investigate the interaction between DCAF10 and Ac-Gly starting peptides, we focused on SFK members. All four tested Nt-acetylated peptides from this family (Lyn, Fyn, Src, and Yes) bound to DCAF10, suggesting a

mechanism relevant to the entire SFK family.

Data are from pull down experiments with modified or unmodified peptides and these data were not recapitulated with intact proteins either recombinant in *in vitro* experiments or from a cellular context (e.g. IPs). Biophysical characterization (e.g. binding assays) for measuring affinities would be desirable. Is the same mechanism applying to all the SFKs members, or are specific and more private determinants for each of the members?

We thank the reviewer for this comment.

First, to establish interaction in a full-length, cellular context, we performed immunoprecipitation of Lyn^{WT}-GFP from cells \pm NMT1/2 siRNA followed by MS to identify interaction partners. These experiments demonstrate association with endogenous DCAF10 in cells; N-terminal mutants do not bind, and NMT1/2 depletion increases DCAF10 association (**Fig. 6j-l**).

Second, regarding biophysical affinities: we attempted equilibrium binding measurements (e.g., SPR) with purified DCAF10, but isolated DCAF10 (without CUL4A-DDB1) proved unstable/aggregation-prone, precluding a reliable K_d assay despite optimization of constructs and buffers. As noted for the myristoylated peptide assays, hydrophobic long-chain effects and nonspecific adsorption also complicate bead-based formats. We therefore focused on feasible assays with streptavidin peptide pull-downs, *in vitro* ubiquitination with reconstituted CUL4A-DDB1-DCAF10, and cellular genetics—which consistently support Ac-Gly-dependent engagement. Future affinity measurements will likely require complex-stabilised DCAF10 preparations.

Third, on generality versus member-specificity within the SFKs: our data indicate shared recognition with distinct magnitudes—Lyn and Fyn show stronger DCAF10 dependence, whereas Src is more sensitive to ZYG11B/ZER1 depletion; combined perturbation is additive, consistent with complementary surveillance branches. To consolidate these differences, all key experiments were repeated as $n = 3$ biological replicates with quantitative reporting (individual data points, mean \pm s.d., fold change; e.g. main **Figs. 2m-p, 4b-f, 4j-m**; Suppl. Figs. *4b-d, 5*).

Together, these additions confirm DCAF10 as a bona fide N-recognin for Ac-Gly-starting SFKs in cellular and biochemical contexts, while delineating member-specific determinants. Detailed quantitative biophysics (K_d) remains an important goal for future work.

4. There are interesting *in silico* prediction of interactions between some modified peptides and the active site of DCAF10. However, there is no functional validation of the key interactive residues that confer specificity and drive such interactions nor any other structure-function relationship experiments. In the same line there is a need to validate such interactions between peptides (both modified and unmodified) and targets biophysically so affinities can be measurable and also provide some more structural evidence about the interacting interface and catalytic reaction.

Could the authors explore further the interaction between c-Src acetylated versus myristoylated and DCAF10 and NatA and map the interaction interfaces and mechanism of regulation?

We thank the reviewer for this insightful comment. We fully agree that detailed structural and biophysical characterization of the interaction interfaces—particularly comparing acetylated versus myristoylated c-Src with DCAF10 and NatA—would provide valuable mechanistic insight. However, full-length biophysical validation was not feasible here because we cannot obtain homogeneous Ac-Gly–modified SFKs (see response above). To establish full-length interaction in cells, we added IPs of Lyn^{WT}–GFP (\pm NMT1/2 RNAi), which show DCAF10 association for WT and loss of binding for N-terminal mutants (new **Fig. 6j–l**). Within the peptide context, we strengthened structure–function mapping by (i) a “swap” experiment (THOC7 acidic segment \leftrightarrow Lyn basic segment, positions ~5–7) showing that acidic residues impede binding, and (ii) “AAA” neutralization mutants in Lyn that maintain DCAF10 binding, together indicating that local context modulates Ac-Gly recognition. (iii) We performed additional AF3 predictions indicating that an Asn4 \rightarrow Ile4 mutation allows deeper Src accommodation in the pocket, consistent with a preference for a hydrophobic residue at position 4. We agree that quantitative affinities (K_d) would be desirable; however, attempts to measure binding using purified DCAF10 were not successful due to instability/aggregation of DCAF10 in isolation. Instead, we focused on streptavidin peptide pull-downs with three independent repeats ($n=3$), reporting percent binding (mean \pm s.d.) to enable comparison across SFKs. Consistent with the modeling, a DCAF10 triple pocket mutant abolishes ubiquitination of Lyn and Fyn but not Src (**Fig. 5i–k**), further supporting the predicted Ac-Gly binding pocket. In sum, while high-resolution biophysics awaits homogeneous Ac-Gly proteins, the added full-length IPs, peptide-based structure–function tests (swap/AAA), quantitative pull-downs, and the DCAF10 pocket mutant together provide a coherent mechanistic picture of DCAF10 recognition of Ac-Gly N-termini within the SFK family.

Across SFKs, DCAF10 binding and dependence differ: Lyn and Fyn bind most strongly, and their ubiquitination is abolished by the DCAF10 pocket mutant, whereas Src binds ~30% less in pull-downs and its ubiquitination is not dependent on the deep pocket (e.g., **Fig. 2m–p**, **Fig. 5i–k**). In cells, we likewise observe member-specific reductions and fold-change patterns upon NMT1/2 depletion and E3 perturbation; Lyn/Fyn show greater DCAF10 dependence, while Src is more sensitive to ZYG11B/ZER1, with additive stabilization when both branches are perturbed (**Fig. 4**). We have expanded the Discussion to acknowledge these shared trends and differences and to clearly state the scope of our conclusions.

5. In the *in vitro* ubiquitination assays. Did the author look at the targeted lysine residues and ID them by MS? Could this bring any insights into the mechanism of interaction and regulation?

We thank the reviewer for this valuable suggestion. We have not yet mapped the targeted lysine residues by MS. We agree that identifying sites and testing non-ubiquitinatable mutants would yield important mechanistic insight and help assess potential biological roles (e.g., site selection and regulation downstream of Ac-Gly/DCAF10 recognition). We plan to address this in follow-up work. In the present manuscript, we deliberately limit the scope to validating and establishing DCAF10 as the substrate receptor for Ac-Gly–starting SFKs.

6. What is the functional role of acetylated versus myristoylated c-Src and SFKs? Are they functionally

distinct in term of phospho-tyrosine activity?

We agree that understanding the functional consequences of N-terminal acetylation versus myristoylation—particularly regarding kinase activity and localization—is among the most relevant questions and a key next step in this project. As an initial indication, we performed a preliminary experiment using full-length Lyn constructs (Lyn^{WT}-GFP, Lyn^{G2A}-GFP, and Lyn^{G2P}-GFP) with or without NMT1/2 RNAi and probed the activating phospho-tyrosine site of Lyn (pY397). We observed marked reduction in phosphorylation for both N-terminal mutants and an approximately 15% decrease for Lyn^{WT} upon NMT1/2 knockdown specifically at the activating pY397 site, while global pY levels appeared to be largely unchanged in these unstimulated cells, while global pY levels seemed to be largely unchanged in these unstimulated cells (**Figure 2** provided to reviewers only). Because these data are exploratory, we have not included them in the manuscript but provide them for the reviewers' information only, as we are working on consequences for kinase activity and biological role. We plan to pursue a systematic analysis under stimulated conditions, quantifying pY patterns and localization changes to define how the N-terminal state affects SFK signaling output. The present study focuses on establishing DCAF10 as the E3 ligase recognizing Ac-Gly–starting SFKs, providing a mechanistic foundation for such future functional work.

Fig. 2 for reviewers only. a) Lyn^{WT}-GFP, Lyn^{G2A}-GFP and Lyn^{G2P}-GFP expressing cells were treated with NMT1/2 RNAi or left untreated and then blotted against anti-pY397 Lyn, pan-anti-pY and controls. b) Quantification of three independent blots.

7. What is known about N-degron pathways in cancer and other human disorders? What is the therapeutic potential of this PTM quality control system? How could it be used therapeutically? Perhaps the author should make emphasis in the discussion on these questions.

We agree with the broader context. While we have not added all these points to the manuscript, we note for the reviewer that (i) UBR1 (E3 ligase of the classical N-degron pathway) loss or mutations underlie the Johanson–Blizzard syndrome, linking N-degron surveillance to genetic human disease (e.g. Hwang, C.-S.

et al., PLOS ONE, 2011). (ii) Co-translational processing enzymes are relevant in oncology—NataA (NAA10/NAA15) is dysregulated in multiple cancers, and NMT1/2 inhibitors are in clinical evaluation; (e.g., Zhu., Gene, 2024; Ree et al., Exp. Mol. Med., 2018, Kim et al., Cancer Res., 2017, Sangha, R., Investigational New Drugs, 2024) and (iii) many cancers exhibit overactive SFKs, for which membrane anchoring is essential.. Conceptually, partial NMT1/2 inhibition could reduce membrane localization and simultaneously expose Ac-Gly/Gly N-degrons for turnover. We thank the reviewer for prompting us to place our findings within this broader biological and translational context and we added this point to the Discussion.

In sum, we thank the reviewer for highlighting several important open areas and we fully agree that these have to be addressed—including quality-control versus potential biological functions of Ac-Gly SFKs, possible crosstalk between co-translational enzymes (NATs/NMT) and E3 ligases, and deeper biophysical/structural mechanisms. Rigorous resolution of these questions will require substantial method development and timelines beyond the present study. We hope that we convinced the reviewer that the data provided establish DCAF10 as the substrate receptor of the CUL4A–DDB1 complex acting as an Ac-Gly–specific N-recognin for SFKs within the defined scope of this manuscript.

REVIEWER COMMENTS _2

We thank all three reviewers for their constructive and positive evaluation of our revised manuscript. We are grateful for the acknowledgement by Reviewers 1 and 2 that all previous concerns were fully addressed and that the revised version has strengthened the manuscript. We also thank Reviewer 3 for the careful re-assessment and for recognizing the substantial additional biochemical and mechanistic work included in this revision. Below, we provide detailed, point-by-point responses to the remaining questions raised by Reviewer 3.

Reviewer #1 (Remarks to the Author):

All concerns have been addressed appropriately.

Reviewer #2 (Remarks to the Author):

The authors have provided clear and satisfactory responses to all my previous comments. They have added the missing Fyn data, standardized quantifications (n = 3), expanded the mechanistic analyses with full-length IP-MS experiments and a DCAF10 pocket mutant, and addressed all minor editorial points. I am satisfied that they have fully addressed my comments.

The revisions strengthen the manuscript without overextending its scope. I have no remaining concerns and recommend acceptance in its current form.

Reviewer #3 (Remarks to the Author):

Revised manuscript NCOMMS-25-22460A

The authors have presented an improved version of the manuscript that answer most of the questions raised by the reviewers. It is clear the authors have tried to provide a more detailed and deeper biochemical analyses of the interacting proteins and partners and their mechanism of action in the study at a cellular level. I acknowledge also that getting N-terminal modified recombinant proteins (SFks members) is a difficult and challenging task and may not be feasible in the time frame of the revision. I assume this is the reason also why they have chosen the pull-down strategy with modified peptides for MS. Regarding the data presented with the full-length proteins in a cellular context I still have some comments that need to be answered and clarified. Mainly, about the interpretation of data from Figure 4 and 6. Besides I have a couple of technical questions that called my attention as well.

Figure 1. In panel g, author show a clear positive correlation between DCAF10 and acetylated-

Gly and not with myristoylated-Gly peptides of Lyn. However, such experimental data is not recapitulated with the other SFKs family members peptides. The authors show however volcano plots with the interacting profile of DCAF10 in presence of acetylated-Gly or unmodified N-terminal peptides from SFKs members. Could the author comment on this please?

The Myr-Gly versus Ac-Gly comparisons for Src and Fyn were performed exactly as shown for Lyn in Fig. 1g. These datasets are available in **Supplementary Fig. 1i-j**, and **Supplementary Fig. 1k** provides the corresponding raw log₂ intensity values for all three SFKs. As our initial analyses focused on comparing acetylated and unmodified (Nt-free) N-terminal peptides, only the Lyn Myr-Ac dataset was included in the main figure, while the equivalent Myr-Ac comparisons for Src and Fyn were placed in the Supplementary Figures referenced in the text (lines 119-122 of the main manuscript: "Further pull-downs using Nt-acetylated and N-myristoylated versions of Lyn, Fyn, and Src peptides confirmed that DCAF10 selectively recognizes the acetylated form of their N-termini (**Fig. 1g, Supplementary Fig. 1i-k, Supplementary Data 1 and 2, Supplementary Table 1**)". Together, these data confirm the same trend across all SFKs examined: DCAF10 selectively enriches acetylated Gly-starting peptides, whereas myristoylated variants do not bind, demonstrating that the pattern observed for Lyn applies consistently to the broader SFK family.

Figure 2. In panel m-p authors show WBs with GST-DCAF10[aa 120-559] which runs in gel at 100 kDa. This size seems to me rather high, compared with data shown on Figure 5b related to MBP-DCAF10 (full-length??) which runs at the same Mw (103 kDa), having in mind the significant difference in size between both tags used (GST-26kDa and MBP-42kDa). Could the authors comment on this please?

We thank the reviewer for raising this point. The apparent discrepancy is due to the fact that different DCAF10 constructs were used for modelling and for experimentation. As stated in the text and detailed in the Methods, aa 120–559 of DCAF10 were used for the AF3 structural predictions, where the N-terminal 119 residues were omitted because they are predicted to be disordered. In contrast, all experimental data—including the Streptavidin pull-downs shown in **Fig. 2m–p** and the CUL4A–DDB1–RBX1–DCAF10 complex purifications displayed in **Fig. 5b**—were generated using full-length DCAF10 (aa 1–559), either as GST–DCAF10 (for peptide binding assays) or as MBP–DCAF10 within the purified E3 ligase complex.

Accordingly, both proteins shown in **Fig. 2m–p** (GST–DCAF10) and **Fig. 5b** (MBP–DCAF10) represent full-length DCAF10 but carry different N-terminal tags. The expected molecular masses are ~86–87 kDa for GST–DCAF10 (GST ≈ 26 kDa) and ~102–103 kDa for MBP–DCAF10 (MBP ≈ 42 kDa). In our gels, GST–DCAF10 runs slightly below 100 kDa, whereas MBP–DCAF10 runs slightly above it, which is reasonable given their mass difference and the resolving power of the SDS–PAGE systems used.

For additional clarity, we now explicitly indicate “full-length” when referring to GST–DCAF10 in the pull-down experiments the first time and when first mentioning the purified E3 ligase complex in the main text.

Besides, it is not clear what are the interpretation of the WBs from panel m-p in terms of expected bands and interactions. Could the authors comment on this please?

The Western blots in **Fig. 2m–p** illustrate the direct binding assay used to validate the MS-based peptide pull-down results. The initial MS experiments showed enrichment of DCAF10 on acetylated Gly-starting peptides, but MS-based enrichment does not, on its own, confirm direct interaction, as additional proteins in the lysate may contribute. To address this, we performed an *in vitro* assay using only the minimal components: purified full-length GST–DCAF10 as bait and streptavidin beads fully loaded with biotinylated N-terminal peptides (following the manufacturer’s recommendation to ensure complete bead saturation).

In this assay, the expected bands are: (i) GST–DCAF10 (or GST alone) in the input, (ii) GST–DCAF10 retained on peptide-loaded beads only when binding occurs, and (iii) streptavidin confirming equal peptide loading. The blots show that acetylated N-terminal peptides of Lyn, Fyn and Src bind robustly and directly to GST–DCAF10, whereas GST alone does not, demonstrating binding specificity. The side-by-side blots and their quantification further show that acetylated peptides are strongly preferred over Nt-free variants, and that Lyn and Fyn peptides bind more strongly than Src, consistent with AlphaFold 3 (AF3) predictions (**Supplementary Figure 4b-d**).

In addition, the authors show docking data however no MD simulation (300-400 ns) is provided to test the stability and thermodynamic parameters of the interaction. I think the peptide should be shown on sticks and the interactions depicted explicitly. These data is important and will add support to the 3D-docking prediction from AF3.

Molecular dynamics (MD) simulations, as suggested, can in principle provide extended -timescale information on the stability and energetics of predicted interfaces. However, MD analyses are not typically required to support AF3-based peptide–protein predictions. AF3 already incorporates extensive physical priors and reliably provides high-confidence local geometry for short linear motifs bound to structured domains. The request for 300–400 ns MD simulations and explicit depiction of all inter-residue contacts corresponds to a level of computational refinement generally used in specialised structural -modelling studies rather than in mechanistic cell-biology work of this scope.

In our study, AF3 serves as a hypothesis-generating tool to propose a plausible binding mode and to rationalize the experimentally observed sequence preferences. We deliberately chose an intuitive mode of representation to convey the key point to a broad audience: acetylated peptides penetrate deeply into the DCAF10 pocket, whereas non-acetylated variants remain at the surface. This behaviour is illustrated clearly in the current depictions, and the key interacting amino acids are shown in stick representation in **Fig. 5i** and **Supplementary Fig. 2l**. Most importantly, the predicted binding behaviour is supported by multiple orthogonal biochemical experiments, including MS-based pull-downs, direct peptide binding assays with purified DCAF10, and DCAF10-dependent *in vitro* ubiquitination. These results independently establish the biological interaction, while the AF3 models provide mechanistic context rather than a fully parameterized thermodynamic description. We hope that this integrated experimental–structural approach is acceptable to the reviewer.

Figure 3. Did the author use N-Myr-peptides as negative controls? Did they also try to monitor enzymatic activity of NatA in presence of DCAF10 towards N-Acetylated SFKs peptides?

In Fig. 3, our aim was to determine whether the different SFK-derived N-termini are competent substrates for NatA-mediated N-terminal acetylation. For this reason, we used the corresponding acetylated SFK peptides as negative controls. Because these represent the final reaction products, they cannot serve as NatA substrates and therefore provide the most direct and stringent control for background signal. As an additional negative control, we included a Pro-starting peptide, which is not a NatA substrate, and we used a published NatA target peptide as a positive control. All experiments were performed in biological quadruplicates and repeated independently, each time with these internal controls, ensuring that the assay is robustly validated.

We did not use N-myristoylated peptides as negative controls, because the myristoyl group blocks the free α -amine and thereby prevents NatA access to the N-terminus. In this context, the acetylated peptide is therefore the most appropriate and biochemically meaningful negative control.

Regarding NatA activity in the presence of DCAF10: this setup was not pursued because DCAF10 binds to the acetylated N-terminus, i.e., the product of the NatA reaction, and has no mechanistic basis to interact with or compete for unmodified substrate peptides. Thus, DCAF10 would not be expected to influence NatA catalytic activity, and for this reason we did not include DCAF10 in the *in vitro* acetylation assays.

Figure 4. Panel a and b. Proposed model 1. N-Myr-Gly protein is viable no degron-mediated degradation. 2. In NMT1/2 deficient cells (depleted) acetylated N-term-Gly SFKs will be degraded via DCAF10 degron. 3. In DCAF10 depleted cells N-Myr-Gly SFKs members, as expected, do not undergo degradation. 4. However, In DCAF10 depleted cells, if NMT1/2 is also depleted N-Acet-Gly proteins will not undergo degradation.

Under these premises, we should expect a drop in the protein levels of all SFKs members under condition 2. However, this is only evident on Lyn and not in c-Src nor Fyn cases. Could the author comment on this please?

In addition, in DCAF10 RNAi cells, I see in some cases that upon depletion of NMT1/2 (hence, favoring N-Acetylated Gly in SFKs) there is a decrease (e.g. Lyn) in some cases and a significant increase (e.g. c-Src) in others, and I do not fully understand these differences as the pattern should be consistent across the distinct SFKs family members. Could the author comment on this please?

It is correct that the steady-state response of individual SFKs to NMT1/2 depletion varies: Lyn shows a pronounced reduction, Fyn a modest but consistent decrease, and Src remains largely unchanged or even increases. Correspondingly, the extent of recovery in the combined NMT1/2 and DCAF10 RNAi condition also differs across SFKs. To further investigate these differences, we analysed SFK stability using cycloheximide and MG132 (**Supplementary Fig. 5d–h**). These experiments showed that Lyn, Fyn, and Src all exhibit increased turnover upon NMT1/2 depletion to a similar degree, demonstrating that Src is affected by the same degradation mechanism even though this is not apparent at the steady-state level. In contrast, the control protein THOC7 shows no changes under any tested condition, confirming that the observed regulation is specific to SFKs. Therefore the quantitative steady-state differences, do not indicate differential pathway engagement. Src behaviour is fully consistent with earlier work showing that Src steady-state abundance can remain buffered (Timms et al., Science 2019).

The varying degrees of reduction and recovery across SFKs likely reflect differences in the cellular balance between translation and degradation for each kinase. Because SFKs are key signalling proteins and the RNAi treatments extend 72 hours, cells have sufficient time to compensate increased degradation by adjusting translation to different extents. Despite these quantitative differences, the qualitative outcome is consistent: all SFKs tested show increased turnover upon NMT1/2 depletion and rescue upon loss of DCAF10, whereas THOC7 remains unaffected. Thus, the apparent differences in steady-state levels arise from kinase-specific degrees of translational compensation and RNAi-related variability, while the underlying turnover behaviour is shared across all examined SFKs.

On panel j, why author show MS data and not WBs or qPCR of DCAF10?

In panel j, MS-based quantification is shown because endogenous DCAF10 could not be detected by Western blot. As stated in the manuscript “DCAF10 was not detectable using multiple anti-DCAF10 antibodies. To overcome this limitation, we employed high-pH fractionation of the whole proteome

combined with MS..." (lines 254-256). During the experimental work, we tested several commercially available antibodies (DCAF10/WDR32), including Thermo Fisher PA5-24133, which we validated to recognize overexpressed DCAF10 in FLAG-DCAF10 transfections and recombinant protein controls. However, none of the antibodies detected endogenous DCAF10 in HeLa cells, most likely due to its very low basal expression level.

To overcome this limitation, we employed whole-proteome mass spectrometry combined with high-pH fractionation, which confirmed both the extremely low endogenous abundance of DCAF10 (ranking >7000 in proteome-wide intensity) and the efficient siRNA-mediated depletion. MS analysis demonstrated a >256-fold reduction in DCAF10 levels after siRNA treatment (**Fig. 4b**). As our study focuses on protein-level regulation, we chose MS rather than qPCR, since proteomic quantification provides a direct and reliable measure of DCAF10 protein abundance and, as a proteomics laboratory, it is our method of choice. The MS corresponding MS data are provided in **Supplementary Data Table 4**.

Figure 5. The new figure 5 is labelled as Figure 4.

We thank the reviewer for drawing our attention to this. The incorrect labelling of the new Fig. 5 as "**Figure 4**" was an oversight on our part, and we have now corrected the figure label.

What does the Ub signal above the 250 kDa band correspond to exactly? Should we expect a prominent band/signal around the size of SFKs (50-60 kDa) which at the same time it will correlate with lower levels of total SFKs members, and viceversa, when Ub is diminished, increased SFKs levels should be seen. If the authors could clarify this.

The pattern observed in the ubiquitin Western blot—a dominant Ub-positive species above 250 kDa (~300 kDa) together with a smear starting at ~150 kDa and extending to the top of the gel—is the typical appearance of polyubiquitinated proteins in *in vitro* ubiquitination assays. Under these conditions, E3 ligases commonly generate long and heterogeneous ubiquitin chains and may modify multiple lysines on the same substrate. This produces a broad high-molecular-weight (high-MW) distribution rather than discrete mono- or multi-ubiquitinated bands near the expected size of the unmodified SFK (~50–60 kDa). Such high-MW smears are widely documented for *in vitro* reactions and reflect the expected heterogeneity of chain lengths.

Because of this intrinsic heterogeneity, the number of attached ubiquitin molecules cannot be inferred from the smear, and changes in the intensity of the unmodified band are typically small and not a reliable measure of ubiquitination efficiency in endpoint assays. In this context, the Ub-positive smear between ~150 kDa and >250 kDa represents the expected readout of polyubiquitin chain formation under the assay conditions used.

Figure 6: On panel a, would the anti-Lyn antibody recognize the ectopic GFP-tagged form of Lyn as well, just as internal control? Why did the authors not show this data?

In **Figure 6a**, all Western blots are probed with anti-Lyn, which detects both endogenous Lyn and the ectopically expressed Lyn–GFP variants. Thus, the GFP-tagged forms are recognized and already shown in this panel. The upper and lower panels derive from the same membrane and correspond to two different exposure times (low and high exposure as written beside the blots): the short exposure visualizes the abundant Lyn–GFP signal, while the long exposure allows detection of endogenous Lyn, which is still not visible at the shorter exposure time and overexpressed Lyn-GFP is already reaching saturation when the endogenous bands appear. For this reason we used two different exposures to visualize both signals on the same blot. All experiments for the cell line were performed with both anti-Lyn and anti-GFP antibodies, and examples are provided in **Supplementary Figure 7d and 7e**. Full low- and high-exposure versions of the blots are included in the **Source Data**.

It would have been nice to see the DLD-1 Lyn KO Flp-In T-REx Lyn-GFP system applied to other SFKs family members.

We agree that extending the DLD-1 Lyn KO Flp-In T-REx system to additional SFK family members would be highly informative. During the revision process, our efforts were directed toward fully addressing the reviewers' first-round comments, and the generation of analogous Fyn and Src cell lines was therefore deprioritized at that stage. Nonetheless, we have already initiated the creation of corresponding Fyn and Src KO and re-expression lines. These models will enable us to compare phosphorylation, localization, and functional similarities and differences across SFKs. These studies are currently ongoing and will be presented in future work.

Figure 7: The new Figure 7 is labelled as Extended Data Figure 8 Kremer et al.

We thank the reviewer for drawing our attention to this. The incorrect labelling of the new **Fig. 7** as “Extended Data Figure 8” was an oversight on our part, and we have now corrected the figure label.

We thank the reviewer for the detailed and thoughtful evaluation of our work, as well as for the constructive suggestions provided during the first revision. We hope that the clarifications and additional explanations supplied here address all remaining questions to the reviewer's satisfaction.